# FedWon: Triumphing Multi-domain Federated Learning Without Normalization

**Weiming Zhuang**
Sony AI
`weiming.zhuang@sony.com`

**Lingjuan Lyu** [*]
Sony AI
`lingjuan.lv@sony.com`

## Abstract

Federated learning (FL) enhances data privacy with collaborative in-situ training on decentralized clients. Nevertheless, FL encounters challenges due to non-independent and identically distributed (non-i.i.d) data, leading to potential performance degradation and hindered convergence. While prior studies predominantly addressed the issue of skewed label distribution, our research addresses a crucial yet frequently overlooked problem known as multi-domain FL. In this scenario, clients' data originate from diverse domains with distinct feature distributions, instead of label distributions. To address the multi-domain problem in FL, we propose a novel method called **Fed**erated learning **With**out **n**ormalizations (FedWon). FedWon draws inspiration from the observation that batch normalization (BN) faces challenges in effectively modeling the statistics of multiple domains, while existing normalization techniques possess their own limitations. In order to address these issues, FedWon eliminates the normalization layers in FL and reparameterizes convolution layers with scaled weight standardization. Through extensive experimentation on five datasets and five models, our comprehensive experimental results demonstrate that FedWon surpasses both FedAvg and the current state-of-the-art method (FedBN) across all experimental setups, achieving notable accuracy improvements of more than 10% in certain domains. Furthermore, FedWon is versatile for both cross-silo and cross-device FL, exhibiting robust domain generalization capability, showcasing strong performance even with a batch size as small as 1, thereby catering to resource-constrained devices. Additionally, FedWon can also effectively tackle the challenge of skewed label distribution.

## 1 Introduction

Federated learning (FL) has emerged as a promising method for distributed machine learning, enabling in-situ model training on decentralized client data. It has been widely adopted in diverse applications, including healthcare (Li et al., 2019; Bernecker et al., 2022), mobile devices (Hard et al., 2018; Paulik et al., 2021), and autonomous vehicles (Zhang et al., 2021; Nguyen et al., 2021; Posner et al., 2021). However, FL commonly suffers from statistical heterogeneity, where the data distributions across clients are non-independent and identically distributed (non-i.i.d) (Li et al., 2020a). This is due to the fact that data generated from different clients is highly likely to have different data distributions, which can cause performance degradation (Zhao et al., 2018; Hsieh et al., 2020; Tan et al., 2023) even divergence in training (Zhuang et al., 2020; 2022b; Wang et al., 2023).

The majority of studies that address the problem of non-i.i.d data focus on the issue of skewed label distribution, where clients have different label distributions (Li et al., 2020b; Hsieh et al., 2020; Wang et al., 2020; Chen et al., 2022). However, multi-domain FL, where clients' data are from different domains, has received less attention, despite its practicality in reality. Figure 1a depicts two practical examples of multi-domain FL. For example, multiple autonomous cars may collaborate on model training, but their data could originate from different weather conditions or times of day, leading to domain gaps in collected images (Cordts et al., 2016; Yu et al., 2020). Similarly, multiple healthcare institutions collaborating on medical imaging analysis may face significant domain gaps due to variations in imaging machines and protocols (Bernecker et al., 2022). Developing effective solutions for multi-domain FL is a critical research problem with broad implications.

---

[*]Corresponding author

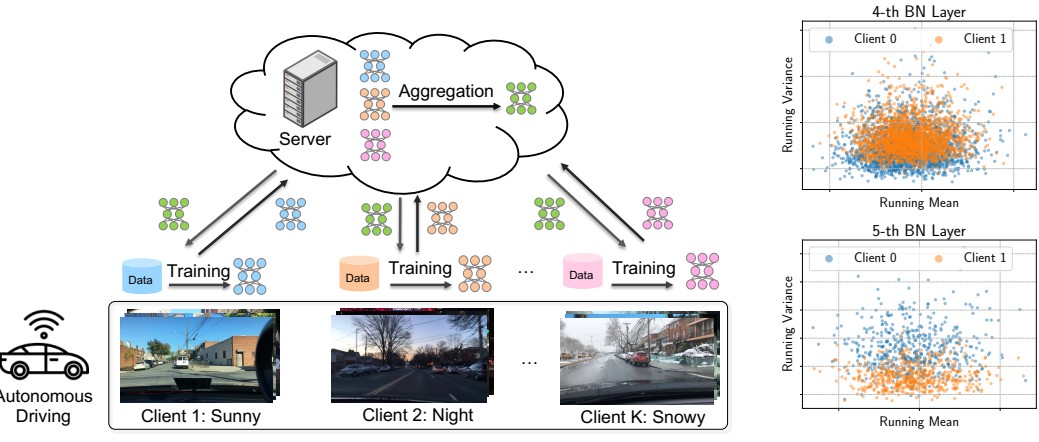

(a) Illustration of Multi-domain FL.  (b) Statistics of BN Layers.

Figure 1: (a) We consider multi-domain federated learning, where each client contains data of one domain. This setting is highly practical and applicable in real-world scenarios. For example, autonomous cars in distinct locations capture images in varying weather conditions. (b) Visualization of batch normalization (BN) channel-wise statistics from two clients, each with data from a single domain. The upper and lower figures are results from the 4-th and 5-th BN layers of a 6-layer CNN, respectively. It highlights different feature statistics of BN layers trained on different domains.

However, the existing solutions are unable to adequately address the problem of multi-domain FL. FedBN (Li et al., 2021) attempts to solve this problem by keeping batch normalization (BN) parameters and statistics (Ioffe & Szegedy, 2015) locally in the client, but it is only suitable for cross-silo FL (Kairouz et al., 2021), where clients are organizations like healthcare institutions, because it requires clients to be stateful (Karimireddy et al., 2020) (keeping states of BN information) and participate training in every round. As a result, FedBN is not suitable for cross-device FL, where the clients are stateless and only a fraction of clients participate in training. Besides, BN relies on the assumption that training data are from the same distribution, ensuring the mean and variance of each mini-batch are representative of the entire data distribution (Ioffe & Szegedy, 2015). Figure 1b shows that the running means and variances of BNs differ significantly between two FL clients from different domains, as well as between the server and clients (statistics of all BN layers are in Figure 12 in Appendix). Alternative normalizations like Layer Norm (Ba et al., 2016) and Group Norm (Wu & He, 2018) have not been studied for multi-domain FL, but they have limitations like requiring extra computation in inference.

This paper explores a fundamentally different approach to address multi-domain FL. Given that BN struggles to capture multi-domain data and other normalizations come with their own limitations, we further ask the question: is normalization indispensable to learning a general global model for multi-domain FL? In recent studies, normalization-free ResNets (Brock et al., 2021a) demonstrates comparable performance to standard ResNets(He et al., 2016). Inspired by these findings, we build upon this methodology and explore its untapped potential within the realm of multi-domain FL.

We introduce **Fed**erated learning **W**ith**o**ut **n**ormalizations (FedWon) to address the domain discrepancies among clients in multi-domain FL. FedWon follows FedAvg (McMahan et al., 2017) protocols for server aggregation and client training. Unlike existing methods, FedWon removes normalization layers (e.g., BN layers), and reparameterizes convolution layers with Scaled Weight Standardization (Brock et al., 2021a). We conduct extensive experiments on five datasets using five models. The experimental results indicate that FedWon outperforms state-of-the-art methods on all datasets and models. The *general global model* trained by FedWon can achieve more than 10% improvement on certain domains compared to the *personalized models* from FedBN (Li et al., 2021). Moreover, our empirical evaluation demonstrated three key benefits of FedWon: 1) FedWon is versatile to support both cross-silo and cross-device FL; 2) FedWon achieves competitive performance on small batch sizes (even on a batch size of 1), which is particularly useful for resource-constrained devices; 3) FedWon can also be applied to address the skewed label distribution problem.

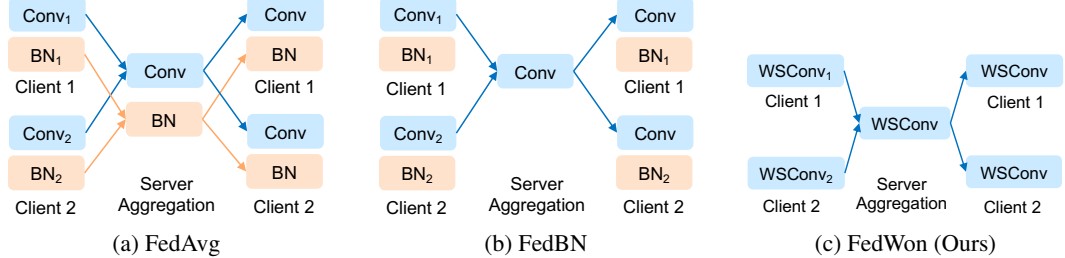

Figure 2: Illustration of three FL algorithms: (a) FedAvg aggregates both convolution (Conv) layers and batch normalization (BN) layers in the server; (b) FedBN keeps BN layers in clients and only aggregates Conv layers; (c) Our proposed **Fed**erated learning **W**ith**o**ut **n**ormalizations (FedWon) removes all BN layers and reparameterizes Conv layers with scaled weight standardization (WSConv).

In summary, our contributions are as follows:

- We introduce FedWon, a simple yet effective method for multi-domain FL. By removing all normalization layers and using scaled weight standardization, FedWon is able to learn a general global model from clients with significant domain discrepancies.

- To the best of our knowledge, FedWon is the first method that enables both cross-silo and cross-device FL without relying on any form of normalization. Our study also reveals the unexplored benefits of this method, particularly in the context of multi-domain FL.

- Extensive experiments demonstrate that FedWon outperforms state-of-the-art methods on all the evaluated datasets and models, and is suitable for training with small batch sizes, which is especially beneficial for cross-device FL in practice.

## 2 PRELIMINARY

Before diving into the benefits brought by removing normalizations, we first introduce FL with batch normalization. Then, we review alternative normalization methods and normalization-free networks.

### 2.1 FEDERATED LEARNING WITH BATCH NORMALIZATION

Batch normalization (BN) (Ioffe & Szegedy, 2015), commonly used as a normalization layer, has been a fundamental component in deep neural networks (DNN). The BN operation is defined as:

$$BN(x) = \gamma \frac{x - \mu}{\sqrt{\sigma^2 + \epsilon}} + \beta, \tag{1}$$

where mean $\mu$ and variance $\sigma$ are computed from a mini-batch of data, and $\gamma$ and $\beta$ are two learnable parameters. The term $\epsilon$ is a small positive value that is added for numerical stability.

BN offers several benefits, including reducing internal covariate shift, stabilizing training, and accelerating convergence (Ioffe & Szegedy, 2015). Moreover, it is more robust to hyperparameters (Bjorck et al., 2018) and has smoother optimization landscapes (Santurkar et al., 2018). However, its effectiveness is based on the assumption that the training data is from the same domain, such that the mean $\mu$ and variance $\sigma$ computed from a mini-batch are representative of the training data (Ioffe & Szegedy, 2015). In centralized training, BN has been found to struggle with modeling the statistics from multiple domains, leading to the development of domain-specific BN techniques (Li et al., 2016; Chang et al., 2019). Similarly, in multi-domain FL, DNN with BN can encounter difficulties in capturing the statistics of multiple domains while training a single global model.

Federated learning (FL) trains machine learning models collaboratively from decentralized clients, coordinated by a central server (Kairouz et al., 2021). It enhances data privacy by keeping the raw data locally on clients. FedAvg (McMahan et al., 2017) is the most popular FL algorithm. A common issue in FL is non-i.i.d data across clients, which could lead to performance degradation and difficulties in convergence (Hsieh et al., 2020; Zhuang et al., 2021; 2022c; Wang et al., 2023).

Skewed label distribution, where clients have different label distributions, is a widely discussed non-i.i.d. problem with numerous proposed solutions (Li et al., 2020b; Zhang et al., 2023; Chen et al., 2022). To address this problem, multiple works provide solutions that introduce special operations on BN to personalize a model for each client (Lu et al., 2022). For example, SiloBN (Andreux et al., 2020) keeps BN statistics locally in clients. FixBN (Zhong et al., 2023) only trains BN statistics in the first stage of training and freezes them thereafter. FedTAN (Wang et al., 2023) tailors for BN by performing iterative layer-wise aggregations, introducing numerous extra communication rounds.

In contrast, multi-domain FL, where the data domains differ across clients, has received less attention (Chen et al., 2018; Shen et al., 2022). FedBN (Li et al., 2021) and FedNorm (Bernecker et al., 2022) addresses this issue by locally keeping the BN layers in clients and aggregating only the other model parameters. PartialFed (Sun et al., 2021) keeps model initialization strategies in clients and use them to load models in new training rounds. While these methods excel in cross-silo FL, where clients are stable and can retain statefulness, they are unsuitable for cross-device FL by design. In the latter scenario, clients are stateless, and only a fraction of clients participate in each round of training (Kairouz et al., 2021). Besides, FMTDA (Yao et al., 2022) adapts source domain data in the server to target domains in clients, whereas we do not assume availablility of data in the server.

## 2.2 Alternative Normalization Methods

BN has shown to be effective in modern DNNs (Ioffe & Szegedy, 2015), but it also has limitations in various scenarios. For example, BN struggles to model statistics of training data from multiple domains (Li et al., 2016; Chang et al., 2019), and it may not be suitable for small batch sizes (Ioffe, 2017; Wu & He, 2018). Researchers have proposed alternative normalizations such as Group Norm (Wu & He, 2018) and Layer Norm (Ba et al., 2016). Although these methods remove some of the constraints of BN, they come with their own limitations. For example, they require additional computation during inference, making them less practical for edge deployment.

Recent studies have shown that BN may not work well in FL under non-i.i.d data (Hsieh et al., 2020), due to external covariate shift (Du et al., 2022) and mismatch between local and global statistics (Wang et al., 2023). Instead, researchers have adopted alternative normalizations such as GN (Hsieh et al., 2020; Casella et al., 2023) or LN (Du et al., 2022; Casella et al., 2023) to mitigate the problem. However, these methods inherit the limitations of GN and LN in centralized training, and the recent study by Zhong et al. (2023) shows that BN and GN have no consistent winner in FL.

## 2.3 Normalization-free Networks

Several attempts have been made to remove normalization from DNNs in centralized training using weight initialization methods (Hanin & Rolnick, 2018; Zhang et al., 2019; De & Smith, 2020). Recently, Brock et al. (2021a) proposed a normalization-free network by analyzing the signal propagation through the forward pass of the network. Normalization-free network stabilizes training with scaled weight standardization that reparameterizes the convolution layer to prevent the mean shift in the hidden activations (Brock et al., 2021a). This approach achieves competitive performance compared to networks with BN on ResNet (He et al., 2016) and EfficientNet (Tan & Le, 2019). Building on this work, Brock et al. further introduced an adaptive gradient clipping (AGC) method that enables training normalization-free networks with large batch sizes (Brock et al., 2021b).

## 3 Federated Learning Without Normalization

In this section, we present the problem setup of multi-domain FL and propose FL without normalization to address the problem of multi-domain FL.

## 3.1 Problem Setup

The standard federated learning aims to train a model with parameters $\theta$ collaboratively from total $N \in \mathbb{N}$ decentralized clients. The goal is to optimize the following problem:

$$\min_{\theta \in \mathbb{R}^d} f(\theta) := \sum_{k=1}^{K} p_k f_k(\theta) := \sum_{k=1}^{K} p_k \mathbb{E}_{\xi_k \sim \mathcal{D}_k}[f_k(\theta; \xi_k)], \tag{2}$$

where $K \in \mathbb{N}$ is the number of participated clients ($K \leq N$), $f_k(\theta)$ is the loss function of client $k$, $p_k$ is the weight for model aggregation in the server, and $\xi_k$ is the data sampled from distribution $\mathcal{D}_k$ of client $k$. FedAvg (McMahan et al., 2017) sets $p_k$ to be proportional to the data size of client $k$. Each client trains for $E \in \mathbb{N}$ local epochs before communicating with the server.

Assume there are $N$ clients in FL and each client $k$ contains $n_k \in \mathbb{N}$ data samples $\{(x_i^k, y_i^k)\}_{i=1}^{n_k}$. Skewed label distribution refers to the scenario where data in clients have different label distributions, i.e. the marginal distributions $\mathcal{P}_k(y)$ may differ across clients ($\mathcal{P}_k(y) \nsim \mathcal{P}_{k'}(y)$ for different clients $k$ and $k'$). In contrast, this work focuses on multi-domain FL, where clients possess data from various domains, and data samples within a client belong to the same domain (Kairouz et al., 2021; Li et al., 2021). Specifically, the marginal distribution $\mathcal{P}_k(x)$ may vary across clients ($\mathcal{P}_k(x) \nsim \mathcal{P}_{k'}(x)$ for different clients $k$ and $k'$). Within each client, the data samples, represented as $x_i$ and $x_j$, drawn from the same marginal distribution $\mathcal{P}_k(x)$ holds that $\mathcal{P}_k(x_i) \sim \mathcal{P}_k(x_j)$ for all $i, j \in 1, 2, ..., n_k$. Figure 1a illustrates practical examples of multi-domain FL. For example, autonomous cars in different locations could capture images under different weather conditions.

## 3.2 Normalization-Free Federated Learning

Figure 1b demonstrates that the BN statistics of clients with data from distinct domains are considerably dissimilar in multi-domain FL. Although various existing approaches have attempted to address this challenge by manipulating or replacing the BN layers with other normalization layers (Li et al., 2021; Du et al., 2022; Zhong et al., 2023), they come with their own set of limitations, such as additional computation cost during inference. To bridge this gap, we propose a novel approach called **Fed**erated learning **W**ith**o**ut **n**ormalizations (FedWon)that removes all normalization layers in FL.

However, simply removing all normalization layers would lead to deteriorated performance in FL. Figure 3 compares the performance of training in a single dataset (SingleSet) and FedAvg without normalization on four domains of the Office-Caltech-10 dataset (Further details on the experimental setup are provided in Section 4). FedAvg without (w/o) BN yields inferior results compared to SingleSet w/o BN. The domain gaps among clients could amplify the challenges in FL when training without BNs.

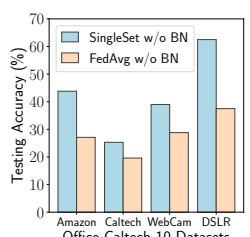

Figure 3: FedAvg without (w/o) BN yields inferior results.

Compared with FedAvg (McMahan et al., 2017), our proposed FedWon completely removes the normalization layers in DNNs and further reparameterizes the convolutions layer. We employ the Scaled Weight Standardization technique proposed by Brock et al. (2021a) to reparameterize the convolution layers after removing BN. The reparameterization formula can be expressed as follows:

$$\hat{W}_{i,j} = \gamma \frac{W_{i,j} - \mu_i}{\sigma_i \sqrt{N}}, \tag{3}$$

where $W_{i,j}$ is the weight matrix of a convolution layer with $i$ as the output channel and $j$ as the input channel, $\gamma$ is a constant number, $N$ is the fan-in of convolution layer, $\mu_i = (1/N)\sum_j W_{i,j}$ and $\sigma_i^2 = (1/N)\sum_j (W_{i,j} - \mu_i)$ are the mean and variance of the $i$-th row of $W_{i,j}$, respectively. By removing normalization layers, FedWon eliminates batch dependency, resolves discrepancies between training and inference, and does not require computation for normalization statistics in inference. We term this parameterized convolution as WSConv.

Figure 2 highlights the algorithmic differences between our proposed FedWon and the other two FL algorithms: FedAvg (McMahan et al., 2017) and FedBN (Li et al., 2021). FedAvg aggregates both convolution and BN layers on the server; FedBN only aggregates the convolution layers and keeps BN layers locally in clients. Unlike these two methods, FedWon removes BN layers, replaces convolution layers with WSConv, and only aggregates these reparameterized convolution layers. Prior work theoretically shows that BN slows down and biases the FL convergence (Wang et al., 2023). FedWon circumvents these issues by removing BN while preserving the convergence speed that BN typically facilitates. Furthermore, FedWon offers unexplored benefits to multi-domain FL, including versatility for both cross-silo and cross-device FL, enhanced domain generalization, and compelling performance on small batch sizes, including a batch size as small as 1.

Table 1: Testing accuracy (%) comparison of different methods on three datasets. Our proposed FedWon outperforms existing methods in most of the domains. FedWon achieves the best average testing accuracy in all datasets.

| | Domains | SingleSet | FedAvg | FedProx | +GN [a] | +LN [b] | SiloBN | FixBN | FedBN | Ours |
|---|---|---|---|---|---|---|---|---|---|---|
| **Digit-Five** | MNIST | 94.4 | 96.2 | 96.4 | 96.4 | 96.4 | 96.2 | 96.3 | 96.5 | **96.8** |
| | SVHN | 67.1 | 71.6 | 71.0 | 76.9 | 75.2 | 71.3 | 71.3 | 77.3 | **77.4** |
| | USPS | 95.4 | 96.3 | 96.1 | 96.6 | 96.4 | 96.0 | 96.1 | 96.9 | **97.0** |
| | SynthDigits | 80.3 | 86.0 | 85.9 | 86.6 | 85.6 | 86.0 | 85.8 | 86.8 | **87.6** |
| | MNIST-M | 77.0 | 82.5 | 83.1 | 83.7 | 82.2 | 83.1 | 83.0 | **84.6** | 84.0 |
| | Average | 83.1 | 86.5 | 86.5 | 88.0 | 87.1 | 86.5 | 86.5 | 88.4 | **88.5** |
| **Caltech-10** | Amazon | 54.5 | 61.8 | 59.9 | 60.8 | 55.0 | 60.8 | 59.2 | **67.2** | 67.0 |
| | Caltech | 40.2 | 44.9 | 44.0 | 50.8 | 41.3 | 44.4 | 44.0 | 45.3 | **50.4** |
| | DSLR | 81.3 | 77.1 | 76.0 | 88.5 | 79.2 | 76.0 | 79.2 | 85.4 | **95.3** |
| | Webcam | 89.3 | 81.4 | 80.8 | 83.6 | 71.8 | 81.9 | 79.6 | 87.5 | **90.7** |
| | Average | 66.3 | 66.3 | 65.2 | 70.9 | 61.8 | 65.8 | 65.5 | 71.4 | **75.6** |
| **DomainNet** | Clipart | 42.7 | 48.9 | 51.1 | 45.4 | 42.7 | 51.8 | 49.2 | 49.9 | **57.2** |
| | Infograph | 24.0 | 26.5 | 24.1 | 21.1 | 23.6 | 25.0 | 24.5 | 28.1 | **28.1** |
| | Painting | 34.2 | 37.7 | 37.3 | 35.4 | 35.3 | 36.4 | 38.2 | 40.4 | **43.7** |
| | Quickdraw | **71.6** | 44.5 | 46.1 | 57.2 | 46.0 | 45.9 | 46.3 | 69.0 | 69.2 |
| | Real | 51.2 | 46.8 | 45.5 | 50.7 | 43.9 | 47.7 | 46.2 | 55.2 | **56.5** |
| | Sketch | 33.5 | 35.7 | 37.5 | 36.5 | 28.9 | 38.0 | 37.4 | 38.2 | **51.9** |
| | Average | 42.9 | 40.0 | 40.2 | 41.1 | 36.7 | 40.8 | 40.3 | 46.8 | **51.1** |

[a]+GN means FedAvg+GN, [b]+LN means FedAvg+LN

## 4 Experiments on Multi-domain FL

In this section, we start by introducing the experimental setup for multi-domain FL. We then validate that FedWon outperforms existing methods in both cross-silo and cross-device FL and achieves comparable performance even with a batch size of 1. We end by providing ablation studies.

### 4.1 Experiment Setup

**Datasets.**  We conduct experiments for multi-domain FL using three datasets: Digits-Five (Li et al., 2021), Office-Caltech-10 (Gong et al., 2012), and DomainNet (Peng et al., 2019). Digits-Five consists of five sets of 28x28 digit images, including MNIST (LeCun et al., 1998), SVHN (Netzer et al., 2011), USPS (Hull, 1994), SynthDigits (Ganin & Lempitsky, 2015), MNIST-M (Ganin & Lempitsky, 2015); each digit dataset represents a domain. Office-Caltech-10 consists of real-world object images from four domains: three domains (WebCam, DSLR, and Amazon) from Office-31 dataset (Saenko et al., 2010) and one domain (Caltech) from Caltech-256 dataset (Griffin et al., 2007). DomainNet (Peng et al., 2019) contains large-sized 244x244 object images in six domains: Clipart, Infograph, Painting, Quickdraw, Real, and Sketch. To mimic the realistic scenarios where clients may not collect a large volume of data, we use a subset of standard digits datasets (7,438 training samples for each dataset instead of tens of thousands) as adopted in Li et al. (2021). We evenly split samples of each dataset into 20 clients for cross-device FL with a total of 100 clients. Similarly, we tailor the DomainNet dataset to include only 10 classes of 2,000-5,000 images. To simulate multi-domain FL, we construct a client to contain images from a single domain.

**Implementation Details.**  We implement FedWon using PyTorch (Paszke et al., 2017) and run experiments on a cluster of eight NVIDIA T4 GPUs. We evaluate the algorithms using three architectures: 6-layer convolution neural network (CNN) (Li et al., 2021) for Digits-Five dataset, AlexNet (Krizhevsky et al., 2017) and ResNet-18 (He et al., 2016) for Office-Caltech-10 dataset, and AlexNet (Krizhevsky et al., 2017) for DomainNet dataset. We use cross-entropy loss and stochastic gradient optimization (SGD) as the optimizer with learning rates tuned over the range of [0.001, 0.1] for all methods. Based on SGD, we adopt adaptive gradient clipping (AGC) that is specially designed for normalization-free networks (Brock et al., 2021b). More details are provided in the supplementary.

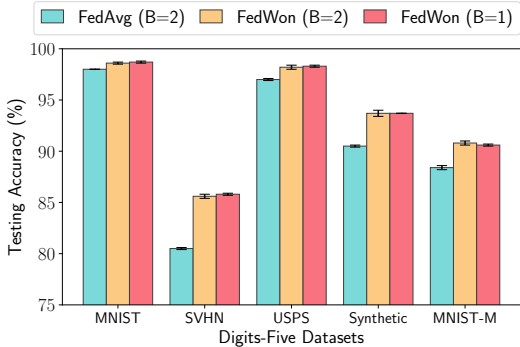 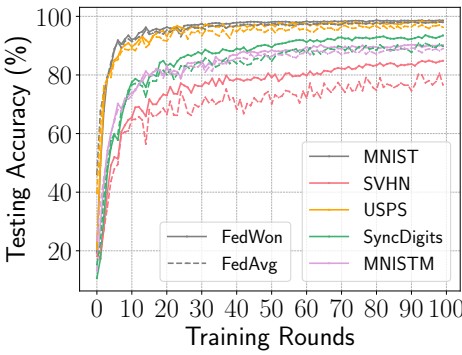

Figure 4: Testing accuracy comparison of FedWon and FedAvg on Digits-Five dataset. Left: comparison of performance using small batch sizes B ={1, 2}, where 10 out of 100 clients are randomly selected to train in each round. Right: comparison of testing accuracy over the course of training with randomly selected 10 out of a total of 100 clients and batch size B = 2.

## 4.2 Performance Evaluation

We compare the performance of our proposed FedWon with the following three types of methods: (1) state-of-the-art methods that employ customized approaches on BN, including SiloBN (Andreux et al., 2020), FedBN (Li et al., 2021), and FixBN (Zhong et al., 2023); (2) baseline algorithms, including FedProx (Li et al., 2020b), FedAvg (McMahan et al., 2017), and SingleSet (i.e. training a model independently in each client with a single dataset); (3) alternative normalization methods, including FedAvg+GN and FedAvg+LN that replace BN layers with GN and LN layers, respectively.

Table 1 presents a comprehensive comparison of the aforementioned methods under cross-silo FL on Digits-Five, Office-Caltech-10, and DomainNet datasets. Our proposed FedWon outperforms the state-of-the-art methods on most of the domains across all datasets. Specifically, FedProx, which adds a proximal term based on FedAvg, performs similarly to FedAvg. These two methods are better than SingleSet in Digits-Five dataset, but they may exhibit inferior performance compared to SingleSet in certain domains on the other two more challenging datasets. SiloBN and FixBN perform similarly to FedAvg, in terms of average accuracy; they are not primarily designed for multi-domain FL and are only capable of achieving the baseline results. In contrast, FedBN is specifically designed to excel in multi-domain FL and outperforms these methods.

Besides, we discover that simply replacing BN with GN (FedAvg+GN) can boost the performance of FedAvg as GN does not depend on the batch statistics specific to domains; FedAvg+GN achieves comparable results as FedBN on Digits-Five and Office-Caltech-10 datasets. Notably, our proposed FedWon surpasses both FedAvg+GN and FedBN in terms of the average accuracy on all datasets. Although FedWon falls slightly behind FedBN by less than 1% on two domains across these datasets, it outperforms FedBN by more than 17% on certain domains. These results demonstrate the effectiveness of FedWon under the cross-silo FL scenario. We report the mean of three runs of experiments here and results with standard deviation in Table 22 in the Appendix.

**Effectiveness on Small Batch Size.** Table 2 compares the performance of our proposed FedWon with state-of-the-art methods using small batch sizes $B = \{1, 2\}$ on Office-Caltech-10 dataset.

Table 2: Performance comparison using small batch sizes $B = \{1, 2\}$ on Office-Caltech-10 dataset. The abbreviations A, C, D, and W respectively represent 4 domains: Amazon, Caltech, DSLR, and WebCam. Our proposed Fed-Won achieves outstanding performance compared to existing methods.

| B | Methods | A | C | D | W |
|---|---------|------|------|------|------|
| 1 | FedAvg+GN | 60.4 | 52.0 | 87.5 | 84.8 |
|   | FedAvg+LN | 55.7 | 43.1 | 84.4 | 88.1 |
|   | **FedWon** | **66.7** | **55.1** | **96.9** | **89.8** |
| 2 | FedAvg | 64.1 | 49.3 | 87.5 | 89.8 |
|   | FedAvg+GN | 63.5 | 52.0 | 81.3 | 84.8 |
|   | FedAvg+LN | 58.3 | 44.9 | 87.5 | 86.4 |
|   | FixBN | 66.2 | 50.7 | 87.5 | 88.1 |
|   | SiloBN | 61.5 | 47.1 | 87.5 | 86.4 |
|   | FedBN | 59.4 | 48.0 | 96.9 | 86.4 |
|   | **FedWon** | **66.2** | **54.7** | **93.8** | **89.8** |

FedWon achieves outstanding performance, with competitive results even at a batch size of 1. While

Table 3: Testing accuracy comparison on randomly selecting a fraction $C = \{0.1, 0.2\}$ out of a total of 100 clients for training each round with batch size $B = 4$ on Digits-Five dataset. FedWon consistently outperforms FedAvg. We report the mean (standard deviation) of three runs of experiments.

| C | Method | MNIST | SVHN | USPS | SynthDigits | MNIST-M | Average |
|---|--------|-------|------|------|-------------|---------|---------|
| 0.1 | FedAvg | 98.2 (0.4) | 81.0 (0.7) | 97.2 (0.5) | 91.6 (1.6) | 89.3 (0.5) | 91.5 (0.8) |
| | **FedWon (Ours)** | **98.6 (0.1)** | **85.4 (0.3)** | **98.3 (0.2)** | **93.6 (0.2)** | **90.5 (0.3)** | **93.3 (0.1)** |
| 0.2 | FedAvg | 97.9 (0.1) | 80.2 (0.0) | 97.0 (0.1) | 91.2 (0.0) | 89.3 (0.0) | 91.1 (0.0) |
| | **FedWon (Ours)** | **98.7 (0.1)** | **86.0 (0.3)** | **98.2 (0.2)** | **94.1 (0.2)** | **90.8 (0.1)** | **93.6 (0.1)** |

FedAvg+GN and FedAvg+LN also achieve comparable results on batch size $B = 1$, they require additional computational cost during inference to calculate the running mean and variance, whereas our method does not have such constraints and achieves even better performance. The capability of our method to perform well under small batch sizes is particularly important for cross-device FL, as some edge devices may only be capable of training with small batch sizes under constrained resources. We have fine-tuned the learning rates for all methods and reported the best ones.

**Cross-device FL with Small Batch Size and Client Selection.** We assess the impact of randomly selecting a fraction of clients to participate in training in each round, which is common in cross-device FL where not all clients join in training. We conduct experiments with fraction $C = \{0.1, 0.2\}$ out of 100 clients on Digits-Five dataset, i.e., $K = \{10, 20\}$ clients are selected to participate in training in each round. Table 3 shows that the performance of our FedWon is better than FedAvg under all client fractions. FedBN is not compared as it is not applicable in cross-device FL. We also evaluate small batch sizes in cross-device FL, with $K = 10$ clients selected per round. Figure 4 (left) shows that the performance of FedAvg degrades with batch size $B = 2$, while our proposed FedWon with batch sizes $B = \{1, 2\}$ achieves consistently comparable results to running with larger batch sizes. Besides, Figure 4 (right) shows the changes in testing accuracy over the course of training. It indicates that FedWon achieves better convergence speed without BN.

**Visualization and Analysis of Feature Maps.** We aim to further study the reason behind the superior performance of FedWon. Figure 5 (top) visualizes feature maps of the last convolution layer of two client local models and one server global model on the Office-Caltech-10 dataset. The feature maps of FedAvg without (w/o) BN have a limited focus on the object of interest. While FedAvg and FedBN perform better, their feature maps display noticeable disparities between client local models.

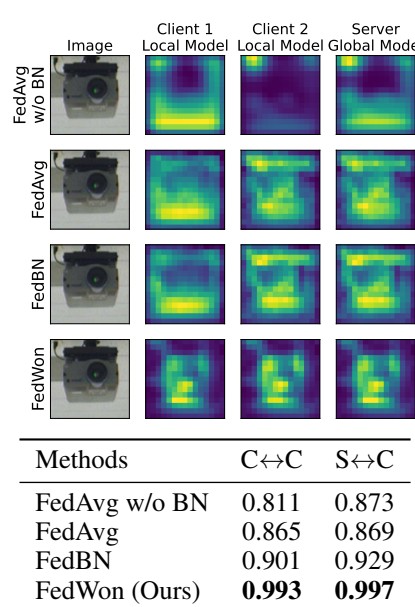

| Methods | C↔C | S↔C |
|---------|------|------|
| FedAvg w/o BN | 0.811 | 0.873 |
| FedAvg | 0.865 | 0.869 |
| FedBN | 0.901 | 0.929 |
| FedWon (Ours) | **0.993** | **0.997** |

Figure 5: Analysis of feature maps with the Caltech-10 dataset. Top: visualization of feature maps of the last convolution layer. Bottom: comparison on average cosine similarity of feature maps between client (C↔C), and between a client and server (S↔C).

In contrast, FedWon showcases superior feature map visualizations, with subtle differences observed among feature maps from different models. To provide further insight, we present the average cosine similarity of all feature maps between client local models (C↔C) and between the server global model and a client local model (S↔C) in Figure 5 (bottom). These results demonstrate the effectiveness of FedWon, which achieves high similarity scores, approaching the maximum value of 1. This finding suggests that FedWon excels at effectively mitigating domain shifts across different domains. Building upon these insights, we extend our analysis to demonstrate that FedWon exhibits superior domain adaptation and generalization capabilities empirically in Table 12 in the Appendix.

We also demonstrate that FedWon achieves significantly superior performance on medical diagnosis in Appendix B.1, which is encouraging and shows the potential of FedWon in the healthcare field.

## 4.3 ABLATION STUDIES

We conduct ablation studies to further analyze the impact of WSConv at batch sizes $B = 32$ and $B = 2$ on the Office-Caltech-10 dataset. Table 4 compares the performance with and without WSConv after removing all normalization layers. It demonstrates that replacing convolution layers with WSConv significantly enhances performance. These experiments use a learning rate of $\eta = 0.08$ for $B = 32$ and $\eta = 0.01$ for $B = 2$. We provide more experiment details in the Appendix B.

Table 4: Ablation studies on the impact of WSConv on Caltech-10 dataset. It significantly boosts performance on both batch sizes B = 32 and B = 2.

| B | WSConv | A | C | D | W |
|---|--------|------|------|------|------|
| 32 | ✓ | **63.7** | **51.0** | **96.3** | **91.2** |
|    |   | 46.4 | 37.3 | 68.8 | 71.2 |
| 2 | ✓ | **67.2** | **55.6** | **96.9** | **93.2** |
|   |   | 54.7 | 44.0 | 84.4 | 78.0 |

## 5 EXPERIMENTS ON SKEWED LABEL DISTRIBUTION

This section extends evaluation from multi-domain FL to skewed label distribution. We demonstrate that our proposed FedWon is also effective in addressing this problem.

**Dataset and Implementation.** We simulate skewed label distribution using CIFAR-10 dataset (Krizhevsky et al., 2009), which comprises 50,000 training samples and 10,000 testing samples. We split training samples into 100 clients and construct i.i.d data and three different levels of label skewness using Dirichlet process Dir($\alpha$) with $\alpha = \{0.1, 0.5, 1\}$, where Dir(0.1) is the most heterogeneous setting. We run experiments using MobileNetV2 (Sandler et al., 2018) with a fraction $C = 0.1$ randomly selected clients (i.e., $K = 10$) out of a total of 100 clients in each round.

| Methods | i.i.d | Dir (1) | Dir (0.5) | Dir (0.1) |
|---------|-------|---------|-----------|-----------|
| FedAvg | 75.0 | 64.5 | 61.1 | 36.0 |
| FedAvg+GN | 65.3 | 58.8 | 51.8 | 21.5 |
| FedAvg+LN | 69.2 | 61.8 | 57.9 | 23.3 |
| FixBN | 75.4 | 64.1 | 61.2 | 34.7 |
| **FedWon (Ours)** | **75.7** | **72.8** | **70.7** | **41.9** |

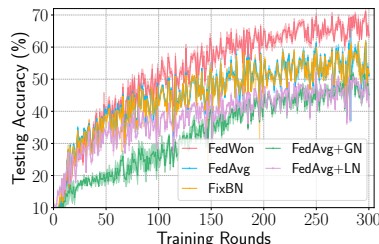

Figure 6: Testing accuracy comparison using MobileNetV2 as backbone on CIFAR-10 dataset. *Left*: performance on different levels of label skewness, where Dir (0.1) represents the most skewed label distribution setting. *Right*: changes in testing accuracy over the course of training on Dir (0.5).

**Performance Comparison.** Figure 6 (left) compares FedWon with FedAvg, FedAvg+GN, FedAvg+LN, and FixBN. FedWon achieves similar performance as FedAvg and FixBN on the i.i.d setting, but outperforms all methods across different degrees of label skewness. We do not compare with FedBN and SiloBN as they are not suitable for cross-device FL and provide the comparison of cross-silo FL scenario in Table 15 in the Appendix. Figure 6 (right) shows changes in testing accuracy over the course of training under the Dir (0.5) setting. FedWon converges to a better position than the other methods. These experiments indicate the possibility of employing our proposed FL without normalization to solve the skewed label distribution problem.

## 6 CONCLUSION

In conclusion, we propose FedWon, a new method for multi-domain FL by removing BN layers from DNNs and reparameterizing convolution layers with weight scaled convolution. Extensive experiments across four datasets and models demonstrate that this simple yet effective method outperforms state-of-the-art methods in a wide range of settings. Notably, FedWon is versatile for both cross-silo and cross-device FL. Its ability to train on small batch sizes is particularly useful for resource-constrained devices. Future work can conduct evaluations of this method under a broader range of datasets and backbones for skewed label distribution. Extending this paradigm from supervised to semi-supervised and unsupervised scenarios is also of interest.

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

## A  EXPERIMENTAL SETUP

In this section, we provide more details of experimental setups, including datasets, model architectures, and implementation details.

### A.1  DATAETS

Figure 7, 8, and 9 visualize three multi-domain datasets used in this work; these three datasets are Digits-Five (Li et al., 2021), Office-Caltech-10 (Gong et al., 2012), and DomainNet (Peng et al., 2019), respectively. It shows that images under each dataset have significant domain gaps. We construct multi-domain FL by constraining each FL client to contain samples of the same domain. Each image is a sample from one client. Each FL client contains images of a dataset (domain). We follow FedBN (Li et al., 2021) to preprocess and transform these datasets.

### A.2  MODEL ARCHITECTURES

Table 5 illustrates the model architectures for experiments on the Digits-Five dataset and Table 6 illustrates the model architectures for experiments on Office-Caltech-10 and DomainNet datasets. For the convolution layer (Conv2D), the hyperparameters are in the sequence of input dimension, output dimension, kernel size, stride, and padding. For the max pooling layer (MaxPool2D), the hyperparameters are kernel and stride. For the fully connected layer (FC), the hyperparameters are input and output dimensions. For the batch normalization (BN) layer, the hyperparameter is the number of channels. For group normalization, the hyperparameters are the number of groups and the number of channels. FedAvg+LN shares a similar model architecture as FedAvg+GN but sets the number of groups to 1. The methods with BN are Standalone, FedAvg (McMahan et al., 2017), FedProx (Li et al., 2020b), SiloBN (Andreux et al., 2020), FedBN (Li et al., 2021), and FixBN (Zhong et al., 2023). These methods share the same model architecture. Note that the model architecture is not exactly the same as the ones used in FedBN (Li et al., 2021), where they use a one-dimension BN layer as regularizer between FC layers but we use Dropout such that the comparisons are fair in terms of model architectures.

Besides, we use the default implementation of ResNet-18 (He et al., 2016) and MobileNetV2 (Paszke et al., 2017) in PyTorch (Paszke et al., 2017) for methods with BN on the Office-Caltech-10 dataset and CIFAR-10 dataset, respectively. FedWon replaces the convolution layers in ResNet-18 and MobileNetV2 with WSConv and removes all batch normalization layers. FedAvg+GN and FedAvg+LN replace BN layers with GN layers. Specifically, FedAvg+GN sets the number of groups to 32 by default, but sets it to 8 when the output dimension is smaller than 32, and to 24 when the output dimension is 144 (to ensure divisibility); FedAvg+LN sets the number of groups in GN to 1. The source code will be released.

### A.3  IMPLEMENTATION AND TRAINING DETAILS

Listing 1 provides the implementation of WSConv in PyTorch. We employ the architectures described in Section A.2 to implement FedWon, adhering to the client training and server aggregation protocols of FedAvg (McMahan et al., 2017). We implement FedWon based on both EasyFL (Zhuang et al., 2022a) for skewed label distribution experiments and FedBN original implementation for multi-domain FL experiments. For the implementation of FedBN, we reference the open-source code available in Github [1]. To implement SiloBN (Andreux et al., 2020), we modify the FedBN implementation to aggregate only the BN parameters while keeping the BN statistics local. Unfortunately, as the source code for FixBN (Zhong et al., 2023) is not publicly available, we implement it based on the description provided in the paper.

Besides, we summarize the compared algorithms in Table 7.

---

[1]https://github.com/med-air/FedBN



| (a) MNIST | (b) SVHN | (c) USPS | (d) SynthDigits | (e) MNIST-M |

Figure 7: Visualization of samples from Digits-Five dataset.

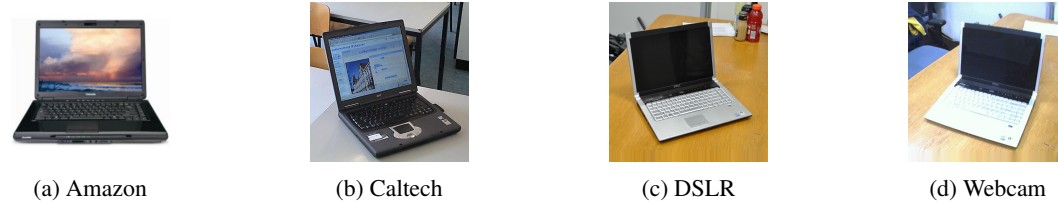

| (a) Amazon | (b) Caltech | (c) DSLR | (d) Webcam |

Figure 8: Visualization of samples from Office-Caltech-10 dataset.

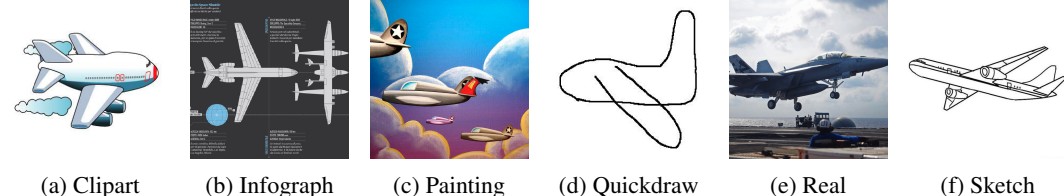

| (a) Clipart | (b) Infograph | (c) Painting | (d) Quickdraw | (e) Real | (f) Sketch |

Figure 9: Visualization of samples from DomainNet dataset.

```python
class WSConv(nn.Conv2d):
  def __init__(self,in_channels,out_channels,kernel_size,stride=1,
  padding=0,dilation=1,groups=1,bias=True,padding_mode='zeros'):
    super(WSConv, self).__init__(in_channels,out_channels,kernel_size,
    stride,padding,dilation,groups,bias,padding_mode)
    nn.init.xavier_normal_(self.weight)
    self.gain = nn.Parameter(torch.ones(self.out_channels, 1, 1, 1))
    _eps = torch.tensor(1e-4, requires_grad=False)
    _fan_in = torch.tensor(self.weight.shape[1:].numel(), requires_grad
    =False).type_as(self.weight)
    self.register_buffer('eps', _eps, persistent=False)
    self.register_buffer('fan_in', _fan_in, persistent=False)

  def standardized_weights(self):
    mean = torch.mean(self.weight, axis=[1,2,3], keepdims=True)
    var = torch.var(self.weight, axis=[1,2,3], keepdims=True)
    scale = torch.rsqrt(torch.maximum(var * self.fan_in, self.eps))
    return (self.weight - mean) * scale * self.gain

  def forward(self, x):
    return F.conv2d(
      input=x,
      weight=self.standardized_weights(),
      bias=self.bias,
      stride=self.stride,
      padding=self.padding,
      dilation=self.dilation,
      groups=self.groups
    )
```

Listing 1: WSConv implementation in PyTorch.

Table 5: Model architectures of Six-layer CNN for experiments on Digits-Five dataset.

| Layer | Methods with BN | FedWon | FedAvg+GN |
|---|---|---|---|
| 1 | Conv2D(3, 64, 5, 1, 2) **BN**(64), ReLU MaxPool2D(2, 2) | **WSConv2D**(3, 64, 5, 1, 2) ReLU MaxPool2D(2, 2) | Conv2D(3, 64, 5, 1, 2) **GN**(32, 64), ReLU MaxPool2D(2, 2) |
| 2 | Conv2D(64, 64, 5, 1, 2) **BN**(64), ReLU MaxPool2D(2, 2) | **WSConv2D**(64, 64, 5, 1, 2) ReLU MaxPool2D(2, 2) | Conv2D(64, 64, 5, 1, 2) **GN**(32, 64), ReLU MaxPool2D(2, 2) |
| 3 | Conv2D(64, 128, 5, 1, 2) **BN**(128), ReLU | **WSConv2D**(64, 128, 5, 1, 2) ReLU | Conv2D(64, 128, 5, 1, 2) **GN**(64, 128), ReLU |
| 4 | Dropout, FC(6272, 2048) ReLU | Dropout, FC(6272, 2048) ReLU | Dropout, FC(6272, 2048) ReLU |
| 5 | Dropout, FC(2048, 512) ReLU | Dropout, FC(2048, 512) ReLU | Dropout, FC(2048, 512) ReLU |
| 6 | FC(512, 10) | FC(512, 10) | FC(512, 10) |

Table 6: Model architectures of AlexNet for experiments on Office-Caltech-10 and DomainNet datasets.

| Layer | Methods with BN | FedWon | FedAvg+GN |
|---|---|---|---|
| 1 | Conv2D(3, 64, 11, 4, 2) **BN**(64), ReLU MaxPool2D(3, 2) | **WSConv2D**(3, 64, 11, 4, 2) ReLU MaxPool2D(3, 2) | Conv2D(3, 64, 11, 4, 2) **GN**(32, 64), ReLU MaxPool2D(3, 2) |
| 2 | Conv2D(64, 192, 5, 1, 2) **BN**(192), ReLU MaxPool2D(3, 2) | **WSConv2D**(64, 192, 5, 1, 2) ReLU MaxPool2D(3, 2) | Conv2D(64, 192, 5, 1, 2) **GN**(32, 192), ReLU MaxPool2D(3, 2) |
| 3 | Conv2D(192, 384, 3, 1, 1) **BN**(384), ReLU | Conv2D(192, 384, 3, 1, 1) ReLU | Conv2D(192, 384, 3, 1, 1) **GN**(64,384), ReLU |
| 4 | Conv2D(384, 256, 3, 1, 1) **BN**(256), ReLU | **WSConv2D**(384, 256, 3, 1, 1) ReLU | Conv2D(384, 256, 3, 1, 1) **GN**(64, 256), ReLU |
| 5 | Conv2D(256, 256, 3, 1, 1) **BN**(256), ReLU MaxPool2D(3, 2) | **WSConv2D**(256, 256, 3, 1, 1) ReLU MaxPool2D(3, 2) | Conv2D(256, 256, 3, 1, 1) **GN**(64, 256), ReLU MaxPool2D(3, 2) |
| 6 | AdaptiveAvgPool2D(6, 6) | AdaptiveAvgPool2D(6, 6) | AdaptiveAvgPool2D(6, 6) |
| 7 | Dropout, FC(9216, 4096) ReLU | Dropout, FC(9216, 4096) ReLU | Dropout, FC(9216, 4096) ReLU |
| 8 | Dropout, FC(4096, 4096) ReLU | Dropout, FC(4096, 4096) ReLU | Dropout, FC(4096, 4096) ReLU |
| 9 | FC(4096, 10) | FC(4096, 10) | FC(4096, 10) |

By default, we conduct experiments with local epochs $E = 1$ and batch size $B = 32$ across all datasets. Stochastic gradient optimization (SGD) is used as the optimizer, with learning rates tuned in the range of [0.001, 0.1] for all methods. Specifically, for FedWon experiments with a batch size of $B = 32$, we incorporate adaptive gradient clipping (AGC) (Brock et al., 2021b), which is specifically designed for normalization-free networks. AGC applies gradient clipping to the weight matrix $W^l \in \mathbb{R}^{N \times M}$ of the $l^{th}$ layer, where the gradient $G^l \in \mathbb{R}^{N \times M}$ is clipped with a threshold $\lambda$ before updating the model. The clipping operation for each row $i$ of $G^l$ can be expressed as follows:

$$G_i^l = \begin{cases} \lambda \frac{\|W_i^l\|_F^*}{\|G_i^l\|_F} G_i^l, & \text{if } \frac{\|G_i^l\|_F}{\|W_i^l\|_F^*} > \lambda, \\ G_i^l, & \text{otherwise,} \end{cases} \tag{4}$$

Table 7: Summary of compared methods on different aspects. ✓and × means the method supports and does not support the attribute, respectively. ◯ means that no prior studies are conducted to analyze whether the method supports the attribute.

| Method | Has no BN | Multi-domain FL | Skewed Labeld Distribution | Cross-silo FL | Cross-device FL |
|---|---|---|---|---|---|
| FedAvg | × | ✓ | ✓ | ✓ | ✓ |
| FedAvg+GN | ✓ | ◯ | ✓ | ✓ | ✓ |
| FedAVG+LN | ✓ | ◯ | ✓ | ✓ | ✓ |
| FixBN | × | × | ✓ | ✓ | ✓ |
| SiloBN | × | ◯ | ✓ | ✓ | × |
| FedBN | × | ✓ | ◯ | ✓ | × |
| FedWon | ✓ | ✓ | ✓ | ✓ | ✓ |

where $|| \cdot ||_F$ is the the Frobenius norm, i.e. $||W^l||_F = \sqrt{\sum_i^N \sum_j^M (W_{i,j})^2)}$, $||W_i^l||_F^* = \max(||W_i||_F, \epsilon)$ with default $\epsilon = 1e - 3$. We only use AGC for FedWon with batch size $B = 32$ and bypass AGC on small batch sizes such as $B = \{1, 2, 4\}$. The impact of AGC and the clipping threshold is further analyzed in Section B.

We tune the learning rates for the methods compared in the main manuscript and provide their specific learning rates below. Table 8 illustrates the learning rates of different methods on three datasets, corresponding to the experiments of Table 1 in the main manuscript. We use a clipping threshold of 0.64 for Digits-Five, 1.28 for Office-Caltech-10, and 1.28 for the DomainNet dataset. Additionally, Table 9 presents the learning rates used for experiments with small batch sizes $B = \{1, 2, 4\}$ on Office-Caltech-10 and Digits-Five datasets. Table 10 displays the learning rates used for experiments using ResNet-18 as the backbone. All experiments on Digits-Five are trained for 100 rounds and experiments on Office-Caltech-10 and DomainNet are trained for 300 rounds.

For evaluation of skewed label distribution, all experiments are run with local epoch $E = 5$ for 300 rounds. We use SGD as the optimizer and tune the learning in the range of [0.001, 0.1] for different algorithms.

Table 8: Learning rates of different methods in the experiments of Table 1 in the manuscript.

| Datasets | Standalone | FedAvg | FedProx | $^a$+GN | $^b$+LN | SiloBN | FixBN | FedBN | Ours |
|---|---|---|---|---|---|---|---|---|---|
| Digits-Five | 0.1 | 0.1 | 0.1 | 0.1 | 0.1 | 0.1 | 0.1 | 0.1 | 0.05 |
| Caltech-10 | 0.01 | 0.01 | 0.01 | 0.01 | 0.01 | 0.01 | 0.01 | 0.01 | 0.1 |
| DomainNet | 0.01 | 0.01 | 0.01 | 0.01 | 0.01 | 0.01 | 0.01 | 0.05 | 0.05 |

$^a$+GN means FedAvg+GN, $^b$+LN means FedAvg+LN

Table 9: Learning rates of experiments on small batch sizes. Left: learning rates of experiments of small batch sizes $B = \{1, 2, 4\}$ on Office-Caltech-10 dataset. Right: learning rates of experiments of small batch sizes of randonly selecting a fraction $C = \{0.1, 0.2, 0.4\}$ out of total clients on Digits-Five dataset.

| B | FedAvg | SiloBN | FixBN | FedBN | FedAvg+GN | FedAvg+LN | Ours |
|---|---|---|---|---|---|---|---|
| 1 | - | - | - | - | 0.001 | 0.001 | 0.005 |
| 2 | 0.001 | 0.001 | 0.001 | 0.001 | 0.001 | 0.001 | 0.01 |
| 4 | 0.001 | 0.01 | 0.01 | 0.01 | 0.001 | 0.001 | 0.03 |

| C | B | FedAvg | Ours |
|---|---|---|---|
| 0.1 | 1 | - | 0.01 |
| 0.1 | 2 | 0.005 | 0.01 |
| 0.1 | 4 | 0.01 | 0.04 |
| 0.2 | 4 | 0.01 | 0.04 |
| 0.4 | 4 | 0.01 | 0.04 |

## B  EXPERIMENTS

This section provides more experiment results that provide further insights into the behavior of FedWon and shed light on the effects of different parameters.

Table 10: Learning rates $\eta$ of different methods in the experiments of using ResNet-20 as backbone.

|  | FedAvg | FedAvg+GN | FedAvg+LN | SiloBN | FixBN | FedBN | Ours |
|---|---|---|---|---|---|---|---|
| $\eta$ | 0.1 | 0.03 | 0.01 | 0.05 | 0.03 | 0.03 | 0.1 |

Table 11: Evaluation on Fed-ISIC2019 dataset with medical images from six different centers. Fed-Won outperforms FedAvg and FedBN by a signifcant margin in all domains.

| Methods | Center 1 | Center 2 | Center 3 | Center 4 | Center 5 | Center 6 |
|---|---|---|---|---|---|---|
| FedAvg | 0.40 | 0.21 | 0.37 | 0.42 | 0.39 | 0.43 |
| FedBN | 0.31 | 0.38 | 0.43 | 0.39 | 0.30 | 0.36 |
| **FedWon (Ours)** | **0.46** | **0.43** | **0.52** | **0.56** | **0.40** | **0.59** |

Table 12: Comparison of methods on domain generalization capability using the Office-Caltech-10 dataset, where we employ Amazon, Caltech, and DSLR as the seen domains during training and WebCam as the unseen domain for evaluation.

| Methods | Seen Domains | | | Unseen Domain |
|---|---|---|---|---|
|  | Amazon | Caltech | DSLR | WebCam |
| FedAvg w/o BN | 38.0 | 33.7 | 40.6 | 28.8 |
| FedAvg | 58.9 | 42.7 | 59.4 | 52.5 |
| FedBN | 65.6 | 48.0 | 78.1 | 61.0 |
| **FedWon (Ours)** | **66.1** | **51.6** | **90.6** | **67.8** |

## B.1 EXPERIMENTS ON MEDICAL IMAGES

To further study how our proposed FedWon benefits multi-domain FL in real-world scenarios, we extend evaluation to diagnosis of skin lesions using datasets from ISIC2019 Challenge (Codella et al., 2018; Combalia et al., 2019) and the HAM10000 (Tschandl et al., 2018) dataset. The dataset contains images collected from four hospitals, where one hospital with 3 different imaging technologies. We follow Flamby (Terrail et al., 2022) to construct them as six different centers: BCN, Vidir-molemax, Vidir-modern, Rosendahl, MSK, and Vienna-dias, with each center's images representing a unique domain. In total, the dataset encompasses 23,247 images of skin lesions, including 9930 training samples and 2483 testing samples from BCN; 3163 training samples and 791 testing samples from Vidir-molemax, 2691 training samples and 672 testing samples from Vidir-modern, 1807 training samples and 452 testing samples from Rosendahl, 655 training samples and 164 testing samples from MSK, and 351 training samples and 88 samples from Vienna-dias. In the experimental setup, we simulate the scenarios where multiple healthcare centers collaborate to train a skin lesion diagnosis model, with each client representing a healthcare center. The task is to conduct image classification for 8 different melanoma classes.

We run the experiments using ResNet-18 (He et al., 2016) (without any pre-training) with local epoch $E = 1$ and batch size $B = 64$ for 50 rounds. We use SGD optimizer with learning rate $\eta = 0.005$ for FedAvg and FedWon and $\eta = 0.001$ for FedBN. The learning rate is tuned among $\{0.001, 0.005, 0.01, 0.05\}$. We follow the implementation in Flamby [2] to use a weighted focal loss (Lin et al., 2017) and data augmentations.

Table 11 shows the testing accuracy of FedAvg, FedBN, and our proposed FedWon across the six healthcare center domains. In this challenging setting, FedBN only achieves similar performance to FedAvg. In contrast, FedWon outperforms both FedAvg and FedBN in all domains by a significant margin. The results are inspiring and demonstrates the potential of deploying FedWon to healthcare application scenarios, where data is often scarce, isolated, and spans multiple domains.

---

[2]https://github.com/owkin/FLamby/

Table 13: Testing accuracy (%) comparison using ResNet-20 on Office-Caltech-10 Dataset.

| Methods | Amazon | Caltech | DSLR | WebCam | Avg |
|---|---|---|---|---|---|
| FedAvg | 45.3 | 36.4 | 68.8 | 76.3 | 56.7 |
| FedAvg+GN | 44.3 | 31.1 | 71.9 | 74.6 | 55.5 |
| FedAvg+LN | 34.4 | 26.2 | 59.4 | 44.1 | 41.0 |
| FixBN | 34.9 | 33.8 | 62.5 | 78.0 | 52.3 |
| SiloBN | 40.6 | 29.3 | 59.4 | 81.4 | 52.7 |
| FedBN | 57.3 | 37.3 | 90.6 | **89.8** | 68.8 |
| **FedWon** | **63.0** | **46.7** | **90.6** | 86.4 | **71.7** |

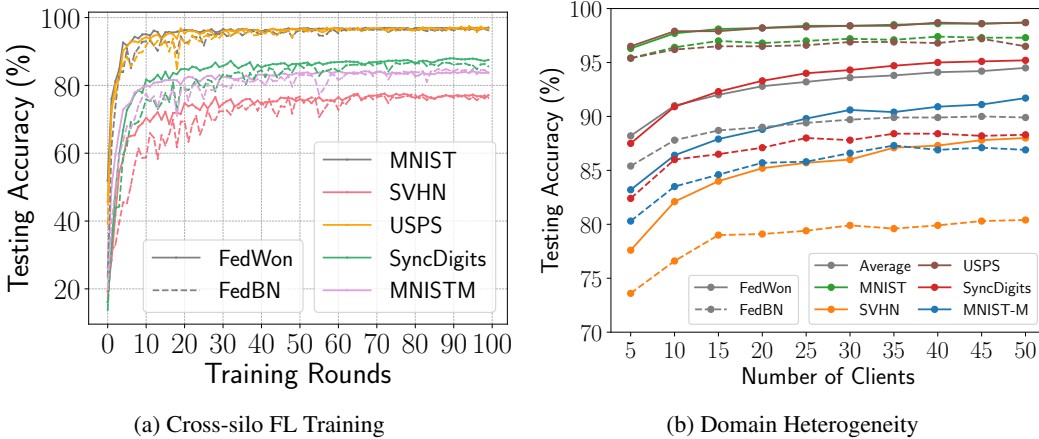

(a) Cross-silo FL Training  (b) Domain Heterogeneity

Figure 10: Testing accuracy (%) comparison of FedBN and FedWon on Digits-Five dataset: (a) compares testing accuracy throughout training on cross-silo FL with total 5 clients (one client per domain) and batch size $B = 32$; (b) comparison of different degrees of domain heterogeneity.

## B.2 ADDITIONAL ANALYSIS

**Domain Generalization Capability.** We expand our analysis to investigate the domain adaptation and generalization capabilities of FedWon. Our experiments are conducted on the Office-Caltech-10 dataset, where we employ Amazon (A), Caltech (C), and DSLR (D) as the seen domains during training, while WebCam (W) is exclusively reserved as an unseen domain only for evaluation, specifically for zero-shot evaluation. We use the client local models to evaluate the seen domains and use the server global model to test on the unseen domain; while to be fair for FedBN, we employ a global model with averaged BN layer parameters from the seen domains. Table 12 presents compelling evidence that FedWon not only excels in performance on the seen domains but also exhibits the most robust generalization capabilities on the unseen domains. These results demonstrate an additional advantage of FedWon, highlighting its capability for domain generalization.

**Evaluation on Alternative Backbones.** In addition to evaluating the effectiveness of FedWon using AlexNet (Krizhevsky et al., 2017) on the Office-Caltech-10 dataset, Table 13 also compares testing accuracy on a common backbone, ResNet-20 (He et al., 2016). Interestingly, replacing BN with GN or LN is not as effective on ResNet-20 as on AlexNet. FedAvg+GN and FedAvg+LN only achieve similar or even worse performance than FedAvg. FedBN (Li et al., 2021), instead, achieves better performance than the other existing methods. Nevertheless, our proposed FedWon consistently outperforms the state-of-the-art methods even with ResNet-20 as the backbone.

**Analysis on Different Degrees of Domain Heterogeneity.** We evaluate the performance of the proposed FedWon under different degrees of domain heterogeneity. To simulate varying degrees of domain heterogeneity, we follow the approach taken by FedBN (Li et al., 2021) and create different numbers of clients with the same domain on the Digits-Five dataset. We start with 5 clients, each containing data from one domain, and then add 5 clients at a time, with each new client containing one of the Digits-Five datasets, respectively. We evaluate the performance of the algorithms for

Table 14: Performance comparison of FedBN and FedWon under different local epochs $E = \{1, 4, 8\}$ on Office-Caltech-10 dataset. FedWon maintains performance and consistently outperforms FedBN under different numbers of local epochs.

| E | Methods | Amazon | Caltech | DSLR | Webcam | Average |
|---|---|---|---|---|---|---|
| 1 | FedBN | 67.2 | 45.3 | 85.4 | 87.5 | 71.4 |
|   | **FedWon** | **67.0** | **50.4** | **95.3** | **90.7** | **75.6** |
| 4 | FedBN | 66.7 | 43.6 | 84.4 | **89.8** | 71.1 |
|   | **FedWon** | **68.8** | **51.1** | **93.8** | 84.8 | **74.6** |
| 8 | FedBN | 64.6 | 45.8 | 87.5 | 89.8 | 71.9 |
|   | **FedWon** | **64.6** | **49.3** | **96.9** | **91.5** | **75.6** |

Table 15: Evaluation on cross-silo skewed label distribution using MobileNetV2 with 10 clients constructed by splitting the CIFAR-10 dataset with Dir (0.1).

| Methods | FedAvg | FedAvg+GN | FedAvg+LN | SiloBN | FixBN | FedBN | FedWon |
|---|---|---|---|---|---|---|---|
| Accuracy | 66.95 | 68.11 | 71.65 | 70.8 | 66.22 | 69.07 | 76.45 |

Table 16: Comparison of FedWon and FedAvg on a total of 1000 clients on Digits-Five dataset, with a selection of only 0.1 clients per round.

| Methods | MNIST | SVHN | USPS | SynthDigits | MNIST-M | Average |
|---|---|---|---|---|---|---|
| FedAvg | 96.0 | 71.2 | 94.7 | 82.9 | **82.8** | 85.5 |
| FedWon | **96.4** | **73.6** | **95.5** | **83.7** | 81.9 | **86.2** |

Table 17: Comparison of FedWon and PartialFed on Office-Caltech-10 dataset.

| Methods | Amazon | Caltech | DSLR | Webcam | Average |
|---|---|---|---|---|---|
| PartialFed-Fix | 58.3 | 44.9 | 88.1 | **91.2** | 70.6 |
| PartialFed-Adaptive | 63.4 | 45.4 | 85.6 | 90.5 | 71.3 |
| FedWon (Ours) | **67.0** | **50.4** | **95.3** | 90.7 | **75.6** |

Table 18: Ablation studies on the impact of WSConv and AGC

| Batch Size | WSConv | AGC | Amazon | Caltech | DSLR | Webcam | Average |
|---|---|---|---|---|---|---|---|
| 32 | ✓ | ✓ | 63.7 | 51.0 | 96.3 | 91.2 | 75.6 |
|    | ✓ |   | 65.1 | 52.0 | 90.6 | 89.8 | 74.4 |
|    |   | ✓ | 46.4 | 37.3 | 68.8 | 71.2 | 55.9 |
|    |   |   | 27.1 | 19.6 | 37.5 | 28.8 | 28.2 |
| 2  | ✓ | ✓ | 65.1 | 51.1 | 93.8 | 86.4 | 74.1 |
|    | ✓ |   | 67.2 | 55.6 | 96.9 | 93.2 | 78.2 |
|    |   | ✓ | 53.1 | 38.7 | 87.5 | 78.0 | 64.3 |
|    |   |   | 54.7 | 44.0 | 84.4 | 78.0 | 65.3 |

different numbers of clients from $N = \{5, 10, 15, ..., 50\}$. More clients represent less heterogeneity as more clients have overlapping domains of data. Figure 10b compares the performance of FedWon and FedBN under these settings. The results show that the performances of both FedWon and FedBN increase as the degree of heterogeneity decreases. FedBN outperforms FedAvg in all the settings as evidenced in Li et al. (2021). However, our proposed FedWon achieves even better performance than FedBN on all domains and all levels of heterogeneity.

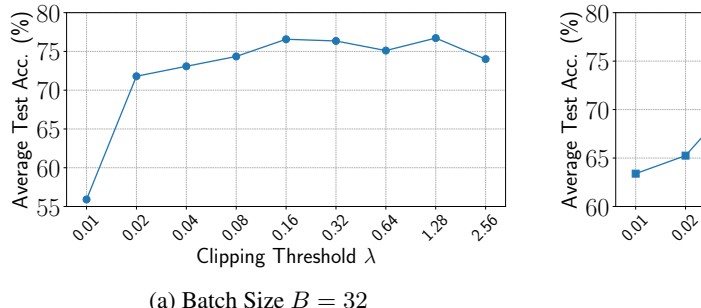
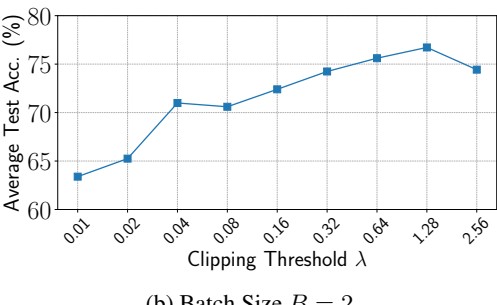

(a) Batch Size $B = 32$          (b) Batch Size $B = 2$

Figure 11: Impact of clipping threshold $\lambda$ of adaptive gradient clipping (AGC) on Office-Caltech-10 dataset, using different batch sizes $B$.

**Testing Accuracy Changes Throughout Training.** Figure 10 illustrates the changes in testing accuracy throughout the training process on the Digits-Five dataset. Specifically, Figure 10a compares the performance of FedWon and FedBN in a cross-silo FL involving a total of 5 clients (one client per domain) and a batch size of $B = 32$. FedWon outperforms FedBN in certain domains or demonstrates similar performance in others. Notably, FedWon achieves better performance in the early stage of training – FedWon exhibits faster convergence, achieving a satisfactory level of accuracy more quickly than FedBN. These results complement the results in Figure 4 (right) in the main manuscript that compares FedWon and FedAvg in a cross-device FL scenario.

**Impact of Local Epochs.** Table 14 compares the performance of our proposed FedWon and FedBN (Li et al., 2021) under different local epochs $E = \{1, 4, 8\}$ on Office-Caltech-10 dataset. Fed-Won maintains performance and consistently outperforms FedBN under different numbers of local epochs. We run these experiments with batch size $B = 32$ and the learning rate the same as the ones in Table 8 on the Office-Caltech-10 dataset.

**Evaluation on Cross-silo FL for Skewed Label Distribution.** Table 15 compares different algorithms in cross-silo FL for skewed label distribution on the CIFAR-10 dataset, which complements cross-device FL experiments in Figure 6. FedWon also consistently outperforms all other methods in cross-silo FL. We run experiments with 10 clients under Dir(0.1) of non-i.i.d data, batch size $B = 64$, and local epoch $E = 5$ for 200 rounds.

**Evaluation on Cross-device FL of 1000 clients.** Table 16 compares FedWon and FedAvg on a total of 1000 clients on Digits-Five dataset, with a selection of only 0.1 clients per round. FedWon also generally outperforms FedAvg under this setting. These experiments are run with batch size B = 2 and learning rate of 0.02.

**Addtional Comparison with PartialFed.** Table 17 further compares FedWon with two variant implementations of PartialFed (Sun et al., 2021) on Office-Caltech-10 dataset. FedWon generally achieves superior performance to PartialFed, especially on the average testing accuracy.

### B.3 ADDITIONAL ABLATION STUDIES

**Impact of WSConv and AGC.** We analyze the impact of WSConv and AGC, which supplements the ablation study presented in the main manuscript. Table 18 shows the impact of these two components with batch size $B = 32$ and small batch size $B = 2$ on the Office-Caltech-10 dataset. After removing the normalizations, using WSConv significantly improves the performance on both batch sizes. AGC, however, shows a positive impact only with batch size $B = 32$, as it is specifically designed for larger batch sizes. Consequently, we do not adopt AGC in the experiments with small batch sizes ($B = \{1, 2, 4\}$). We run these experiments with learning rate $\eta = 0.08$ for $B = 32$ and $\eta = 0.01$ for $B = 2$.

**Impact of Clipping Threshold $\lambda$ for AGC.** We further extend to evaluate the impact of clipping threshold $\lambda$ under batch sizes $B = 2$ and $B = 32$. Figure 11 shows the average testing accuracy on the Office-Caltech-10 dataset using different clipping thresholds $\lambda = \{0.01, 0.02, 0.04, 0.08, 0.16, 0.32, 0.64, 1.28, 2.56\}$. When the batch size $B = 32$, the performance

Table 19: Evaluation on the impact of using AGC optimizer on different algorithms. AGC also benefits other methods, while our proposed FedWon achieves the best overall performance.

| Methods | AGC | Amazon | Caltech | DSLR | Webcam | Average |
|---|---|---|---|---|---|---|
| FedAvg | | 61.8 | 44.9 | 77.1 | 81.4 | 66.3 |
| FedAvg | ✓ | 62.5 | 45.3 | 75.0 | 84.8 | 66.9 |
| FedAvg+GN | | 60.8 | 50.8 | 88.5 | 83.6 | 70.9 |
| FedAvg+GN | ✓ | 64.1 | 48.0 | 90.6 | 88.1 | 72.7 |
| FedAvg+LN | | 55.0 | 41.3 | 79.2 | 71.8 | 61.8 |
| FedAvg+LN | ✓ | 59.4 | 42.2 | 84.4 | 79.7 | 66.4 |
| FixBN | | 59.2 | 44.0 | 79.2 | 79.6 | 65.5 |
| FixBN | ✓ | 58.9 | 43.1 | 75.0 | 88.1 | 66.3 |
| SiloBN | | 60.8 | 44.4 | 76.0 | 81.9 | 65.8 |
| SiloBN | ✓ | 59.4 | 44.4 | 78.1 | 83.0 | 66.2 |
| FedBN | | **67.2** | 45.3 | 85.4 | 87.5 | 71.4 |
| FedBN | ✓ | **70.3** | 45.3 | 87.5 | 88.1 | 72.8 |
| FedWon (Ours) | ✓ | 67.0 | **50.4** | **95.3** | **90.7** | **75.6** |

is rather insensitive to different values of $\lambda$ when it is not too small (larger than 0.08). When the batch size $B = 2$, the best clipping threshold is $\lambda = 1.28$ and the performance is sensitive to different values. Consistent with the finding in Table 18, we recommend avoiding using AGC when the batch size is small. These results provide insights into selecting an appropriate clipping threshold for multi-domain FL.

**Impact of AGC on Other Algorithms.** Table 19 compares the results with and without AGC with the same learning rate for different methods. AGC also benefits other methods, while our proposed FedWon achieves the best overall performance.

### B.4 COMPLEMENTARY EXPERIMENTS

**Visualization of BN Statistics.** Figure 12 visualizes the running mean and variance of BN layers of the 6-layer CNN. It complements Figure 1b in the main manuscript and shows the discrepancies of BN statistics between clients and between a client and the server in all BN layers.

**Visualization of Feature Maps.** Figure 13 presents the visualization of feature maps obtained through three methods: FedAvg, FedBN, and our proposed FedWon. These feature maps are the output of each convolution layer in AlexNet on the Office-Caltech-10 dataset, which encompasses data from four distinct domains, namely Amazon, Caltech, DSLR, and WebCam. FedWon exhibits significantly enhanced feature maps on the object of interest compared to those produced by the FedAvg and FedBN.

**Effectiveness on Small Size.** Table 20 compares performance of our proposed FedWon with existing methods on Office-Caltech-10 dataset with batch size $B = 4$. FedWon achieves the best performance also in this setting, complementing the experiments of batch size $B = \{1, 2\}$ in Table 2 in the main manuscript. Additionally, Figure 14 compares the testing accuracy over the course of training of FedWon with batch sizes $B = \{1, 2, 4\}$ on Digits-Five dataset. Different batch sizes tend to have a similar trend of convergence.

**Effectiveness on Selection of Clients.** Table 21 compares the performance of FedAvg and FedWon on cross-device FL on the Digits-Five dataset with a fraction $C = 0.4$ of clients out of a total of 100 clients to participate in training each round. FedWon achieves superior performance also in this setting, complementing the experiments of $C = \{0.1, 0.2\}$ in Table 3 in the main manuscript.

**Comparison of Methods with Variances.** Table 22 presents testing accuracy comparison of different methods on three datasets with mean (standard deviation) of three runs of experiments. It complements the results in Table 1 in the main manuscript.

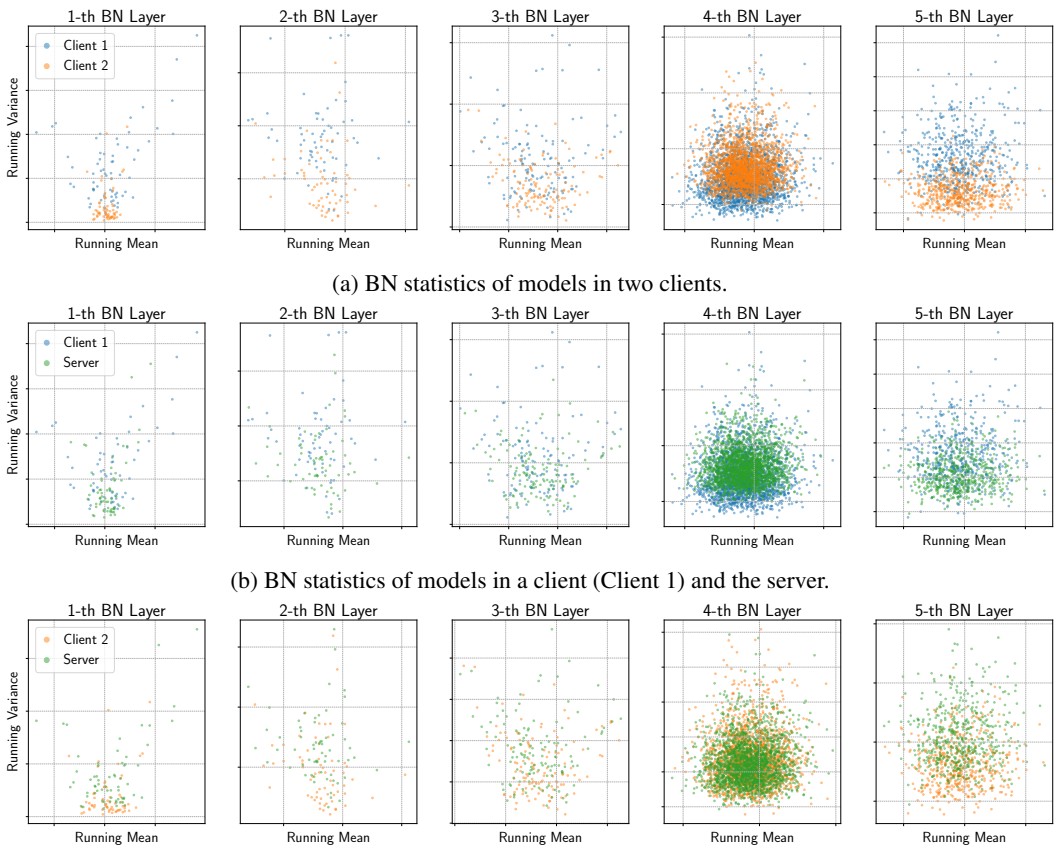

(a) BN statistics of models in two clients.

(b) BN statistics of models in a client (Client 1) and the server.

(c) BN statistics of models in a client (Client 2) and the server.

Figure 12: BN statistics of all layers in a 6-layer CNN.

Table 20: Performance comparison using small batch sizes $B = 4$ on Office-Caltech-10 dataset. Our proposed FedWon achieves outstanding performance compared to existing methods.

| B | Methods | Amazon | Caltech | DSLR | WebCam |
|---|---|---|---|---|---|
| | FedAvg | 65.6 | 46.7 | 78.1 | 88.1 |
| | FedAvg+GN | 60.9 | 52.0 | 84.4 | 89.8 |
| | FedAvg+LN | 54.2 | 44.9 | 78.1 | 72.9 |
| 4 | FixBN | 66.2 | 50.2 | 78.1 | 91.5 |
| | SiloBN | 63.5 | 48.9 | 78.1 | 88.1 |
| | FedBN | 67.2 | 50.7 | 90.6 | 91.5 |
| | **FedWon** | **68.8** | **54.2** | **96.9** | **91.5** |

Table 21: Testing accuracy comparison on randomly selecting a fraction $C = 0.4$ out of a total of 100 clients for training each round with batch size $B = 4$. FedWon consistently outperforms FedAvg on Digits-Five dataset. We report the mean (standard deviation) of three runs of experiments.

| C | Method | MNIST | SVHN | USPS | SynthDigits | MNIST-M | Average |
|---|---|---|---|---|---|---|---|
| 0.4 | FedAvg | 98.1 (0.0) | 80.5 (0.1) | 97.0 (0.2) | 91.4 (0.2) | 89.4 (0.1) | 91.3 (0.0) |
| | **FedWon (Ours)** | **98.8 (0.0)** | **86.4 (0.2)** | **98.4 (0.2)** | **94.2 (0.2)** | **91.0 (0.3)** | **93.7 (0.0)** |

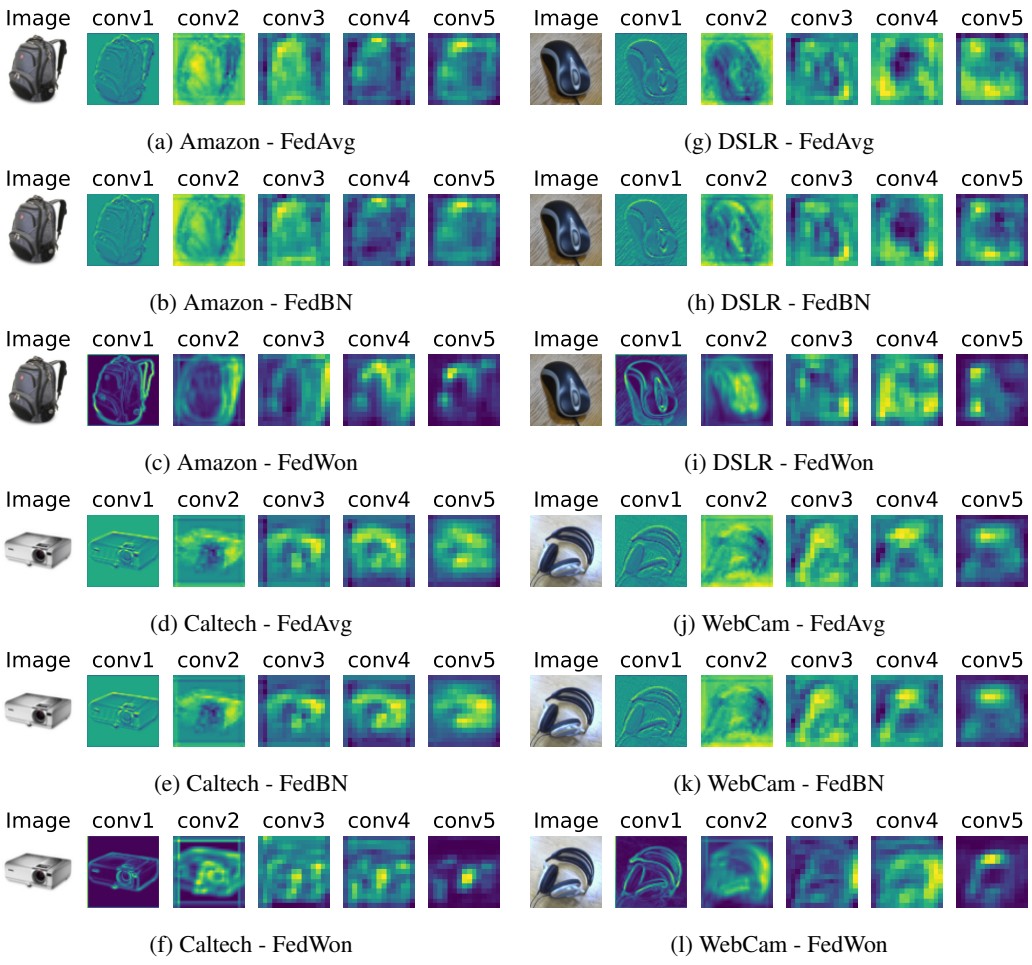

Figure 13: Visualization of feature maps of FedAvg, FedBN, and our proposed FedWon on the Office-Caltech-10 dataset, which contain four domains: Amazon, Caltech, DSLR, and WebCam.

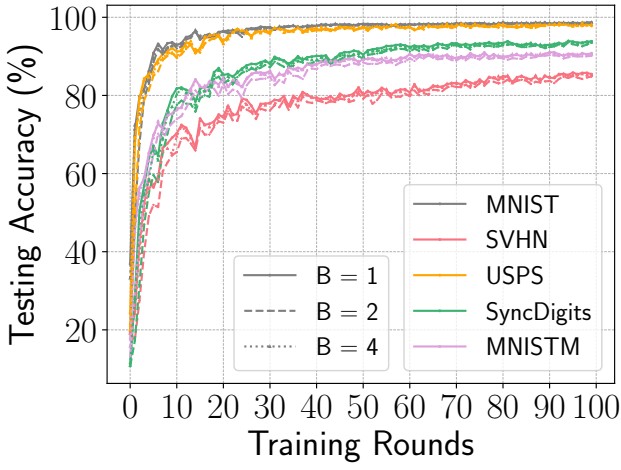

Figure 14: Testing accuracy (%) comparison of different batch sizes $B = \{1, 2, 4\}$ using FedWon on Digits-Five dataset with 10 randomly selected clients out of 100 clients.

Table 22: Testing accuracy (%) comparison of different methods on three datasets. Our proposed FedWon outperforms existing methods in most of the domains. FedWon achieves the best average testing accuracy in all datasets. We report the mean (standard deviation) of three runs of experiments.

| | Domains | Standalone | FedAvg | FedProx | +GN [a] | +LN [b] | SiloBN | FixBN | FedBN | Ours |
|---|---|---|---|---|---|---|---|---|---|---|
| **Digit-Five** | MNIST | 94.4 (0.2) | 96.2 (0.2) | 96.4 (0.0) | 96.4 (0.1) | 96.4 (0.1) | 96.2 (0.0) | 96.3 (0.1) | 96.5 (0.1) | **96.8 (0.2)** |
| | SVHN | 67.1 (0.7) | 71.6 (0.5) | 71.0 (0.8) | 76.9 (0.1) | 75.2 (0.4) | 71.3 (1.0) | 71.3 (0.9) | 77.3 (0.4) | **77.4 (0.1)** |
| | USPS | 95.4 (0.1) | 96.3 (0.3) | 96.1 (0.1) | 96.6 (0.2) | 96.4 (0.4) | 96.0 (0.2) | 96.1 (0.2) | 96.9 (0.2) | **97.0 (0.1)** |
| | SynthDigits | 80.3 (0.8) | 86.0 (0.3) | 85.9 (0.2) | 86.6 (0.1) | 85.6 (0.3) | 86.0 (0.3) | 85.8 (0.1) | 86.8 (0.3) | **87.6 (0.2)** |
| | MNIST-M | 77.0 (0.9) | 82.5 (0.1) | 83.1 (0.2) | 83.7 (0.5) | 82.2 (0.3) | 83.1 (0.4) | 83.0 (0.8) | **84.6 (0.2)** | 84.0 (0.2) |
| | Average | 83.1 (0.4) | 86.5 (0.1) | 86.5 (0.1) | 88.0 (0.1) | 87.1 (0.0) | 86.5 (0.3) | 86.5 (0.0) | 88.4 (0.1) | **88.5 (0.1)** |
| **Office-Caltech-10** | Amazon | 54.5 (1.8) | 61.8 (1.2) | 59.9 (0.5) | 60.8 (1.8) | 55.0 (0.3) | 60.8 (1.3) | 59.2 (1.8) | **67.2 (0.9)** | 67.0 (0.7) |
| | Caltech | 40.2 (0.7) | 44.9 (1.2) | 44.0 (1.9) | 50.8 (3.3) | 41.3 (1.2) | 44.4 (1.2) | 44.0 (0.8) | 45.3 (1.3) | **50.4 (2.8)** |
| | DSLR | 81.3 (0.0) | 77.1 (1.8) | 76.0 (1.8) | 88.5 (1.8) | 79.2 (1.8) | 76.0 (1.8) | 79.2 (1.8) | 85.4 (1.8) | **95.3 (2.2)** |
| | Webcam | 89.3 (1.0) | 81.4 (1.7) | 80.8 (2.6) | 83.6 (5.2) | 71.8 (2.0) | 81.9 (2.0) | 79.6 (2.9) | 87.5 (1.0) | **90.7 (1.2)** |
| | Average | 66.3 (0.4) | 66.3 (0.7) | 65.2 (1.0) | 70.9 (2.5) | 61.8 (0.7) | 65.8 (0.2) | 65.5 (0.8) | 71.4 (1.0) | **75.6 (1.4)** |
| **DomainNet** | Clipart | 42.7 (2.7) | 48.9 (2.0) | 51.1 (0.8) | 45.4 (0.5) | 42.7 (0.7) | 51.8 (1.0) | 49.2 (1.8) | 49.9 (0.5) | **57.2 (0.5)** |
| | Infograph | 24.0 (1.6) | 26.5 (2.5) | 24.1 (1.6) | 21.1 (1.1) | 23.6 (1.2) | 25.0 (2.1) | 24.5 (0.9) | 28.1 (0.8) | **28.1 (0.2)** |
| | Painting | 34.2 (1.6) | 37.7 (3.3) | 37.3 (2.0) | 35.4 (2.0) | 35.3 (0.6) | 36.4 (1.9) | 38.2 (0.7) | 40.4 (0.7) | **43.7 (1.2)** |
| | Quickdraw | **71.6 (0.9)** | 44.5 (3.4) | 46.1 (3.8) | 57.2 (1.0) | 46.0 (1.2) | 45.9 (2.8) | 46.3 (3.9) | 69.0 (0.8) | 69.2 (0.2) |
| | Real | 51.2 (1.0) | 46.8 (2.3) | 45.5 (0.6) | 50.7 (0.3) | 43.9 (0.7) | 47.7 (0.9) | 46.2 (2.8) | 55.2 (2.6) | **56.5 (0.4)** |
| | Sketch | 33.5 (1.1) | 35.7 (0.9) | 37.5 (2.3) | 36.5 (1.8) | 28.9 (1.3) | 38.0 (1.9) | 37.4 (2.0) | 38.2 (6.7) | **51.9 (1.9)** |
| | Average | 42.9 (0.5) | 40.0 (1.5) | 40.2 (0.5) | 41.1 (0.0) | 36.7 (0.3) | 40.8 (0.4) | 40.3 (0.3) | 46.8 (1.5) | **51.1 (0.2)** |

[a] +GN means FedAvg+GN, [b] +LN means FedAvg+LN

