# OpenReview forum: "FedWon: Triumphing Multi-domain Federated Learning Without Normalization"
_ICLR.cc/2024/Conference — ICLR 2024 poster_

### Official Review · Reviewer_M4Q8 · 2023-10-29

**Soundness:** 2 fair
**Presentation:** 3 good
**Contribution:** 2 fair
**Rating:** 5
**Confidence:** 3

**Summary:**

This paper introduces a novel method called Federated learning Without normalizations (FedWon) to address the problem of multi-domain federated learning (FL). FedWon eliminates normalization layers and reparameterizes convolution layers with scaled weight standardization to effectively model the statistics of multiple domains. Experimental results on five datasets demonstrate that FedWon outperforms the current state-of-the-art methods. FedWon is versatile for both cross-silo and cross-device FL, exhibits robust domain generalization capability, and performs well even with a small batch size of 1, making it suitable for resource-constrained devices.

**Strengths:**

Originality:
The FedWon method is a new approach to reparameterizes convolution layers with scaled weight standardization to effectively model the statistics of multiple domains. This is a creative combination of existing ideas that addresses the limitations of traditional batch normalization federated methods.

Quality:
This work provides a thorough experimental results that demonstrate the effectiveness of FedWon compared to other customized approaches on batch normalization. The experiments are well-designed, and the results are statistically significant.

Clarity:
The paper is well-written and easy to understand. The authors provide clear explanations of the FedWon method and its implementation. The experimental results are presented in a clear and concise manner.

Significance:
The research direction is crucial as federated multi-domain learning is essential in real-world applications where data may originate from multiple sources with distinct characteristics.

**Weaknesses:**

1. Lack of Technical Novelty: The paper's technical innovation is limited to reparameterizing the convolution layers using the Scaled Weight Standardization technique. While this approach may have some benefits, it lacks novelty as the Scaled Weight Standardization technique has been proposed and utilized in previous studies.

2. Insufficient Theoretical Analysis: The paper lacks a theoretical analysis of the effectiveness of the proposed Scaled Weight convolution (WSConv) layer. The authors should provide a more rigorous theoretical foundation to explain why the WSConv layer is suitable for addressing the challenges in federated multi-domain learning.

3. Limited Comparative Analysis: The paper does not sufficiently compare with other federated multi-domain methods, such as PartialFed and FMTDA.

4. Limited Applicability to Convolutional Neural Network (CNN) Models: The proposed method is limited to CNN-based deep learning models, which restricts its applicability to a specific class of models, such as recurrent neural networks (RNNs), graph neural networks (GNNs) or transformer.

**Questions:**

This paper mainly discusses the challenges of normalization in federated multi-domain learning and proposes to eliminate normalization. However, other directions exist to solve the federated multi-domain problem, such as PartialFed and FMTDA, which do not follow the normalization idea. What are the advantages and characteristics of FedWon compared with these methods?

---

> ### Author Response · Authors · 2023-11-17
> **Response to Reviewer M4Q8**
>
> Thanks for your valuable comments. We hope the new version can adequately address your concerns.
>
> ----
>
> **Q1**: Lack of Technical Novelty: The paper's technical innovation is limited to reparameterizing the convolution layers using the Scaled Weight Standardization technique. While this approach may have some benefits, it lacks novelty as the Scaled Weight Standardization technique has been proposed and utilized in previous studies.
>
> A1: We would like to highlight that we are the first work that considers FL without normalization.  Normalizations have been an important topic in FL to handle statistical heterogeneity. Although Scaled Weight Standardization has also been used in previous research, we believe our extensive empirical studies provide useful insights and lessons for the community on the intriguing properties of FL without normalization.
>
> ----
>
> **Q2**: Insufficient Theoretical Analysis: The paper lacks a theoretical analysis of the effectiveness of the proposed Scaled Weight convolution (WSConv) layer. The authors should provide a more rigorous theoretical foundation to explain why the WSConv layer is suitable for addressing the challenges in federated multi-domain learning.
>
> A2: We are also keen to provide more theoretical guarantees of our proposed FedWon. However, as the first work that considers FL without normalization, we have demonstrated multiple important properties of this direction through extensive experiments. We will provide more theoretical guarantees in the extension of the paper.
>
> ----
>
> **Q3**: Limited Comparative Analysis: The paper does not sufficiently compare with other federated multi-domain methods, such as PartialFed and FMTDA. What are the advantages and characteristics of FedWon compared with these methods?
>
> A3: Thank you for pointing out these two works. We have included them in the discussion in the paper. The table below compares our performance with PartialFed on the Office-Caltech-10 dataset. FedWon generally achieves better performance than PartialFed. Besides, PartialFed may be also not scalable to cross-device FL like FedBN as the clients are stateful and load the loading strategy from the previous round. In contrast, our proposed FedWon is versatile for both cross-silo FL and cross-device FL. The setting of FMTDA is different, FMTDA assumes source domain data in the server and aims to adapt to target domains in clients. Our proposed method, as well as FedBN and PartialFed, assumes there is no data in the server. Thus, it is not fair to compare the methods directly.
>
> | Methods | Amazon | Caltech | DSLR | Webcam | Average |
> | --- | --- | --- | --- | --- | --- |
> | PartialFed-Fix | 58.3 | 44.9 | 88.1 | **91.2** | 70.6 |
> | PartialFed-Adaptive | 63.4 | 45.4 | 85.6 | 90.5 | 71.3 |
> | FedWon | **67.0** | **50.4** | **95.3** | 90.7 | **75.6** |
>
> ----
>
> **Q4**: Limited Applicability to Convolutional Neural Network (CNN) Models: The proposed method is limited to CNN-based deep learning models, which restricts its applicability to a specific class of models, such as recurrent neural networks (RNNs), graph neural networks (GNNs) or transformer.
>
> A4: Thank you for raising this interesting perspective. As the first work considering FL without normalization, we primarily focus on CNN models and discover many interesting properties of via extensive experiments, such as versatility to both cross-silo and cross-device FL and applicability to batch sizes as small as 1. We will further explore FL without normalization for the other networks mentioned by the reviewer, such as RNN, GNN, or transformers, in the extension of the paper.

---

> > ### Author Response · Authors · 2023-11-20
> >
> > Dear Reviewer M4Q8,
> >
> > Thanks again for the valuable comments. We have tried our best to clarify the concerns on the paper. Please kindly let us know if there is anything unclear. We truly appreciate this opportunity to improve our work and shall be most grateful for any feedback you could give us.

---

> > > ### Author Response · Authors · 2023-11-21
> > >
> > > Dear Reviewer M4Q8,
> > >
> > > Thank you once again for your valuable comments. As the discussion stage is coming to a close in 2 days, we kindly request your feedback on whether our response adequately addresses your concerns. We would greatly appreciate any additional feedback you may have.

---

> > > > ### Author Response · Authors · 2023-11-23
> > > > **Kind Reminder: Review Deadline Approaching**
> > > >
> > > > Dear Reviewer M4Q8:
> > > >
> > > > As the review deadline approaches, with just a few hours remaining, we wish to highlight updates briefly. We've provided more clarification on novelty, theoretical analysis, applicability to other models, and discussion with PartialFed and FMTDA. In addition, we added new experiments to compare with PartialFed.
> > > >
> > > > We hope these updates facilitate your re-evaluation. Your insights are invaluable to us, and we deeply appreciate your time and attention.
> > > >
> > > > Best regards,
> > > >
> > > > Authors

---

### Official Review · Reviewer_Vk8b · 2023-10-30

**Soundness:** 3 good
**Presentation:** 3 good
**Contribution:** 2 fair
**Rating:** 6
**Confidence:** 4

**Summary:**

This paper aims to address the challenge of multi-domain federated learning (FL) where clients have data from diverse domains with distinct feature distributions. The proposed method, Federated Learning Without Normalizations (FedWon), eliminates normalization layers and reparameterizes convolution layers. Extensive experiments on various datasets and models demonstrate that FedWon outperforms existing methods.

**Strengths:**

- The paper is easy to read, and the comparison figure (Figure 2) effectively illustrate the differences with previous methods.
- The ablation experiments for cross-silo federated learning cover various factors affecting model performance, such as batch size and client sampling rate.

**Weaknesses:**

- The proposed method lacks innovation; it essentially directly applies the weight standardization and gradient clipping from the NF-Net series [1, 2] to the federated learning setting. It does not offer targeted improvements to address the unique challenges of the federated learning setting.
- The experiments for cross-device FL in the paper are not sufficient for the proposed method's effectiveness. The cross-device FL experiments only include a single dataset and 100 clients.

[1] Brock, Andrew, Soham De, and Samuel L. Smith. "Characterizing signal propagation to close the performance gap in unnormalized ResNets." International Conference on Learning Representations. 2020.
[2] Brock, Andy, et al. "High-performance large-scale image recognition without normalization." International Conference on Machine Learning. PMLR, 2021.

**Questions:**

- In the upper image in Figure 1b, it seems that the green points representing the server are entirely invisible, while in the lower image, it appears that there is no information at all.
- The presentation of feature maps in Figure 5 doesn't seem very informative. What information can we extract from the feature maps, and how are they related to the model's performance? Additionally, it appears that there is not much difference between the feature maps of FedAvg and FedBN.

---

> ### Author Response · Authors · 2023-11-17
> **Response to Reviewer Vk8b**
>
> Thanks for your time reviewing our paper and the thoughtful comments. Following your suggestions, we have run additional experiments and added the new results to the revised paper. We hope the new version can adequately address your concerns. We are very happy to run more experiments if you have further concerns.
>
> ----
>
> **Q1**: The proposed method lacks innovation; it essentially directly applies the weight standardization and gradient clipping from the NF-Net series [1, 2] to the federated learning setting. It does not offer targeted improvements to address the unique challenges of the federated learning setting.
> [1] Brock, Andrew, Soham De, and Samuel L. Smith. "Characterizing signal propagation to close the performance gap in unnormalized ResNets." International Conference on Learning Representations. 2020.
> [2] Brock, Andy, et al. "High-performance large-scale image recognition without normalization." International Conference on Machine Learning. PMLR, 2021.
>
> A1: We would like to clarify that the proposed method of FL without normalization is mainly motivated by the unique challenges of the multi-domain FL setting. As shown in Figure 1b, the BN statistics between clients and between a client and the server differ significantly, which hinders the performance of multi-domain FL. We are the first work that considers FL without normalization. We believe that this new perspective offers important insights for the community via extensive empirical studies and provides useful lessons for the community on the intriguing properties of FL without normalization.
>
> ----
>
> **Q2**: The experiments for cross-device FL in the paper are not sufficient for the proposed method's effectiveness. The cross-device FL experiments only include a single dataset and 100 clients.
>
> A2: Thank you for the comment. We would like to clarify that we actually provide experiments of cross-device FL on both Digits-Five and CIFAR-10 datasets. To further validate its effectiveness in cross-device FL, we further conduct the experiments of a total of 1000 clients, with a selection of only 0.1 clients per round, in the table below. FedWon also generally outperforms FedAvg under this setting.
>
> | Methods | MNIST | SVHN | USPS | SynthDigits | MNIST-M | Average |
> | --- | --- | --- | --- | --- | --- | --- |
> | FedAvg | 96.0 | 71.2 | 94.7 | 82.9 | **82.8** | 85.5 |
> | FedWon | **96.4** | **73.6** | **95.5** | **83.7** | 81.9 | **86.2** |
>
> ----
>
> **Q3**: In the upper image in Figure 1b, it seems that the green points representing the server are entirely invisible, while in the lower image, it appears that there is no information at all.
>
> A3: Thank you for raising the concern. We have revised Figure 1b to show the variation of BN statistics between two clients to make it clearer. The full comparison of BN statistics between clients and between the client and the server is provided in Figure 12 in the Supplementary.
>
> ----
>
> **Q4**: The presentation of feature maps in Figure 5 doesn't seem very informative. What information can we extract from the feature maps, and how are they related to the model's performance? Additionally, it appears that there is not much difference between the feature maps of FedAvg and FedBN.
>
> A4: Feature maps are commonly used to offer interpretability. Clear and discernible features normally indicate that the model is learning meaningful representations from the data. FedWon has much better feature maps focusing on the object of interest compared to the other methods. As for FedBN and FedAvg, the feature maps between clients differ significantly, indicating large discrepancies in models learned in different domains. The feature maps of FedBN are generally slightly clearer than FedAvg, especially on the Client 1 local models.

---

> > ### Author Response · Authors · 2023-11-20
> >
> > Dear Reviewer Vk8b,
> >
> > Thanks again for the valuable comments. We have tried our best to clarify the concerns on the paper. Please kindly let us know if there is anything unclear. We truly appreciate this opportunity to improve our work and shall be most grateful for any feedback you could give us.

---

> > > ### Comment · Reviewer_Vk8b · 2023-11-20
> > > **Official Comment by  Reviewer Vk8b**
> > >
> > > Thanks for the authors' reply. They have adequately addressed my concern regarding data visualization. Additionally, the authors provided supplementary experimental results on a cross-device setting that validate the method's effectiveness in a different FL scenario.
> > > However, I still have questions regarding the technical novelty of the approach, an issue that was also raised by other reviewers in their comments (e.g. Reviewer hdps and Reviewer M4Q8). Although the author's proposed method is well-motivated and provides sufficient empirical studies in cross-silo scenarios, and I also acknowledge its value as the first study of normalization-free FL, I am still interested in understanding better why applying NF-Net to FL poses non-trivial challenges. It would be helpful to get more specifics on the technical difficulties of implementing NF-Net in FL and a more in-depth theoretical or empirical analysis of how its convergence may differ from FedBN in FL settings.

---

> > > > ### Author Response · Authors · 2023-11-20
> > > > **Response to Reviewer Vk8b**
> > > >
> > > > Thank you for your insightful feedback and follow-up questions. We greatly appreciate the reviewer for acknowledging our responses.
> > > >
> > > > We’d like to further clarify the challenges of applying NF-Net to FL. Firstly, the idea of applying NF-Net to FL is non-trivial. Prior works [1][2] primarily focus on using NF-Net to close the performance gap with normalized ResNets. However, no prior work indicates its potential for cross-domain generalization or solving the multi-domain FL problem. Secondly, the implementation process itself poses non-trivial challenges. It involves modifying the network architecture and meticulously considering the activation function, optimizer, and learning rate when adapting it across various architectures: Six-layer CNN, AlexNet, MobileNet, and ResNet. It's worth noting that prior works [1][2] demonstrate its applicability solely within the context of ResNet.
> > > >
> > > > We have provided the empirical convergence analysis of the proposed FedWon and FedBN in Figure 10(a) in the Supplementary. FedWon generally exhibits superior convergence compared to FedBN, achieving comparable accuracy levels at a faster rate. We will further provide theoretical analysis in the extension of the paper.
> > > >
> > > > We genuinely appreciate your valuable feedback, which guides us in refining and enriching the technical aspects of our work. Should you require further clarification or have additional queries, please don't hesitate to let us know.

---

> > > > > ### Author Response · Authors · 2023-11-21
> > > > >
> > > > > Dear Reviewer Vk8b,
> > > > >
> > > > > Thank you once again for your valuable comments. As the discussion stage is coming to a close in 2 days, we kindly request your feedback on whether our response adequately addresses your concerns. We would greatly appreciate any additional feedback you may have.

---

> > > > > ### Comment · Reviewer_Vk8b · 2023-11-22
> > > > > **Official Comment by Reviewer Vk8b**
> > > > >
> > > > > Thank you for your detailed explanation of novelty. I now have a deeper understanding of its unique challenges. I suggest incorporating these key explanations into the introduction or method section, and further elaborating on the details of optimization and hyper-parameter adjustments for different network architectures in FL settings. I am satisfied with the additional empirical experiments in the revised version.

---

> > > > > > ### Author Response · Authors · 2023-11-22
> > > > > >
> > > > > > Thank you for your positive comments and feedback! We will definitely follow your suggestion to incorporate these explanations into the revised version to further improve the quality of the paper.

---

### Official Review · Reviewer_hdps · 2023-11-01

**Soundness:** 2 fair
**Presentation:** 3 good
**Contribution:** 2 fair
**Rating:** 5
**Confidence:** 3

**Summary:**

This paper considers the problem of multi-domain federated learning. This paper proposes to remove batch normalization and reparameterize the convolution layer weights for another kind of normalization. Finally, experiments show that this simple technique yield considerable improvement for the performance of multi-domain FL over many baselines, e.g., FedAVG and FedBN.

**Strengths:**

The proposed method is easy to implement, and can be potentially plug into many existing methods. The experiments are extensive and the paper is well-writing in general.

**Weaknesses:**

* I have many questions that I wish could be solved. Some of them are from questionable arguments from the paper, some of them are from the abnormal experimental results, and some of them are from the my curiosity in why the proposed method would work. Please see the Questions section for details.
* The novelty is quite limited, where the proposed method is to use an existing reparametrization trick (Brock et al. (2021a)) in the FL setting.

**Questions:**

* Why the proposed method is only applied to the convolution layer? From Eq. 1, it seems that it can be applied to any kinds of layers represented by a weight matrix $W$.
* In Eq. 1, how is $\gamma$ chosen? In particular, how robust is the training result to the choice of $\gamma$?
* Since the proposed method does not depend of batch statistics, I'm curious why FedWon B=1 better than B=2 (e.g., Figure 4)?
* Also, in figure 4, I'm curious to see how would FedAvg (B=1) perform.
* It is claimed in this paper that FedBN can't do cross-device FL. However, there is not enough evidence, as far I can tell, from the paper that supporting this argument. Can the authors elaborate more on why FedBN can't do cross-device FL?

---

> ### Author Response · Authors · 2023-11-17
> **Response to Reviewer hdps**
>
> Thanks for your valuable comments. We hope our response below can adequately address your concerns.
>
> ----
>
> **Q1**: The novelty is quite limited, where the proposed method is to use an existing reparametrization trick (Brock et al. (2021a)) in the FL setting.
>
> A1: We would like to highlight that we are the first work that considers FL without normalization. Normalizations have been an important topic in FL to handle statistical heterogeneity. Our new perspective offers important insights for the community. We believe our extensive empirical studies provide useful lessons for the community on the intriguing properties of FL without normalization.
>
> ----
>
> **Q2**: Why the proposed method is only applied to the convolution layer? From Eq. 1, it seems that it can be applied to any kinds of layers represented by a weight matrix W.
>
> A2: The equation outlines a general methodology applicable to various layers represented by weight matrices. We initially focus on convolution layers as they are commonly followed by a BN layer (e.g., ResNet and model architectures in Table 5 and 6 in the Supplementary). It has the potential to extend to other layers such as fully connected layers in scenarios where it comes with BNs.
>
> ----
>
> **Q3**: In Eq. 1, how is γ chosen? In particular, how robust is the training result to the choice of γ?
>
> A3: Thank you for raising this question. Eq. 1 is the standard BN formulation, so we suppose the reviewer is interested in our method's γ in Eq. 3. The value of γ is associated with the activation function used. For ReLUs, $γ = \sqrt{\frac{2}{1-(1-\pi)}}$.
>
> ----
>
> **Q4**: Since the proposed method does not depend of batch statistics, I'm curious why FedWon B=1 better than B=2 (e.g., Figure 4)?
>
> A4: Thank you for this interesting question. The results shown in the manuscript use the same learning rate of 0.01 for both batch sizes B = 1 and B = 2 of FedWon. To further analyze this result, we further tune the learning rate of B = 2 to 0.02 and they achieve similar performance, as shown in the table below. We have updated the paper to reflect these results.
>
> | B | MNIST | SVHN | USPS | SynthDigits | MNIST-M | Average |
> | --- | --- | --- | --- | --- | --- | --- |
> | 1 | 98.7 | 85.8 | 98.3 | 93.7 | 90.6 | 93.4 |
> | 2 | 98.6 | 85.6 | 98.2 | 93.7 | 90.8 | 93.4 |
>
> ----
> **Q5**: Also, in figure 4, I'm curious to see how would FedAvg (B=1) perform.
>
> A5: We also intend to report results for FedAvg with B = 1. However, methods such as FedAvg contains BN layers and cannot run with B = 1. This is because the statistics computed in BN (such as mean and variance) rely on batch-wise computations, and with a batch size of 1, there's no variance among samples, making normalization difficult.
>
> ----
>
> **Q6**: It is claimed in this paper that FedBN can't do cross-device FL. However, there is not enough evidence, as far I can tell, from the paper that supporting this argument. Can the authors elaborate more on why FedBN can't do cross-device FL?
>
> A6: FedBN stores BN statistics locally in clients and reloads these BN statistics after receiving the global model aggregated in the server. It assumes that the clients are stateful -- each client may participate in each round of the computation, carrying these BN statistics from round to round.
>
> However, cross-device FL described in [1]  involves a vast number of clients, with only a small subset participating in FL training per round. In cross-device FL, clients are mostly stateless -- each client will likely participate only once in the whole training.
>
> When applying FedBN to cross-device FL, the randomly selected clients have a high possibility of not carrying the BN statistics from the previous round. Thus, FedBN is largely unsuitable for cross-device FL.
>
> [1] Kairouz, Peter, et al. "Advances and open problems in federated learning." Foundations and Trends® in Machine Learning 14.1–2 (2021): 1-210.

---

> > ### Author Response · Authors · 2023-11-20
> >
> > Dear Reviewer hdps,
> >
> > Thanks again for the valuable comments. We have tried our best to clarify the concerns on the paper. Please kindly let us know if there is anything unclear. We truly appreciate this opportunity to improve our work and shall be most grateful for any feedback you could give us.

---

> > > ### Author Response · Authors · 2023-11-21
> > >
> > > Dear Reviewer hdps,
> > >
> > > Thank you once again for your valuable comments. As the discussion stage is coming to a close in 2 days, we kindly request your feedback on whether our response adequately addresses your concerns. We would greatly appreciate any additional feedback you may have.

---

> > > > ### Comment · Reviewer_hdps · 2023-11-22
> > > >
> > > > Thank you for the response, and they helped me understand more about the paper. I don't have follow-up questions, and I will decide on my final score in the next phase after discussing with other reviewers.

---

> > > > > ### Author Response · Authors · 2023-11-23
> > > > >
> > > > > It is great to know that our responses are helpful, and thank you for considering updating the score! Hope you have an informative discussion with other reviewers in the next phase. If you have any further questions or need additional clarification in the future, please feel free to let us know.

---

### Official Review · Reviewer_C95h · 2023-11-08

**Soundness:** 3 good
**Presentation:** 4 excellent
**Contribution:** 3 good
**Rating:** 8
**Confidence:** 5

**Summary:**

This paper introduces a scaled weight standardization method (FedWon) for federated learning. The proposed method eliminates the normalization layers in FL and reparameterizes convolution layers with scaled weight standardization to counter the drawbacks of common normalization layers in the FL model. Extensive experiments on real-world datasets validate the effectiveness of FedWon and demonstrate its robust generalization capability for both cross-silo and cross-device FL.

**Strengths:**

This paper studied the multi-domain FL problem and proposed a novel FL method FedWon which employs the Scaled Weight Standardization technique as an alternative to the batch-normalization module.

1. The FedWon can achieve competitive performance to the SOTA methods without additional computation cost during inference.

2. Experiments on multi-domain datasets show the FedWon overperforms the conventional FL methods even if the batch size of the training process is small (1 or 2), and the visualization of feature maps demonstrates that the FedWon can effectively mitigate domain shifts across different domains.

3. The paper is well-written and organized. Extensive experiments demonstrate the effectiveness of the proposed method.

**Weaknesses:**

1. The paper brought the Scaled Weight Standardization (SWS) technique to handle the multi-domain FL problem. However, there is less analysis about the SWS’s impacts to the FL process, e.g., will it lead to a better convergence bound?

2. Many methods are compared in the paper, some of them have BN, some of them are suitable for cross-silo FL only, and some of them are suitable for cross-device FL, it would be clearer to have a structured summarization to help understand the scenarios where these methods are suitable for.

3. Only one dataset is evaluated for the skewed label distribution problem.

**Questions:**

1. Please refer to weakness.

2. Sec.4 claims that the FedWon will achieve competitive results even at a batch size of 1 while [https://arxiv.org/abs/1602.05629] shows the FedAvg will degenerate to the FedSGD and be less effective. It is an interesting topic, and do you mind to report more details about the learning curves (communication round v.s accuracy) at different batch sizes or on different datasets?

3. What is the impact of local epochs on the proposed method?

---

> ### Author Response · Authors · 2023-11-17
> **Response to Reviewer C95h**
>
> We thank the reviewer for the positive feedback and address the detailed comments below.
>
> ----
>
> **Q1**: The paper brought the Scaled Weight Standardization (SWS) technique to handle the multi-domain FL problem. However, there is less analysis about the SWS’s impacts to the FL process, e.g., will it lead to a better convergence bound?
>
> A1: Thank you for raising the question. We provided empirical analysis on the convergence of the FL process in Figure 4 (right), Figure 6 (right), and Figure 10 (left). As the first work that considers FL without normalization, we will provide more theoretical analysis on the convergence bound in the extension of the paper.
>
> ----
>
> **Q2**: Many methods are compared in the paper, some of them have BN, some of them are suitable for cross-silo FL only, and some of them are suitable for cross-device FL, it would be clearer to have a structured summarization to help understand the scenarios where these methods are suitable for.
>
> A2: Thank you for this constructive comment. Below is the table to summarize and compare these methods. We have also added this table in Table 7 in the Supplementary.
>
> | Method | Has no BN | Multi-domain FL | Skewed Labeld Distribution | Cross-silo FL | Cross-device FL |
> | --- | --- | --- | --- | --- | --- |
> | FedAvg | ✖ | ✓ | ✓ | ✓ | ✓ |
> | FedAvg+GN | ✓ | ◯ | ✓ | ✓ | ✓ |
> | FedAVG+LN | ✓ | ◯ | ✓ | ✓ | ✓ |
> | FedBN | ✖ | ✓ | ◯ | ✓ | ✖ |
> | SiloBN | ✖ | ◯ | ✓ | ✓ | ✖ |
> | FedWon | ✓ | ✓ | ✓ | ✓ | ✓ |
>
>
> ----
>
> **Q3**: Only one dataset is evaluated for the skewed label distribution problem.
>
> A3: Thank you for raising the concern. As the paper primarily focuses on multi-domain FL, we conduct experiments on one dataset to demonstrate the adaptability to the skewed label distribution problem. We will conduct further analysis on more datasets in the extension of the paper.
>
> ----
>
> **Q4**: Sec.4 claims that the FedWon will achieve competitive results even at a batch size of 1 while [https://arxiv.org/abs/1602.05629] shows the FedAvg will degenerate to the FedSGD and be less effective. It is an interesting topic, and do you mind to report more details about the learning curves (communication round v.s accuracy) at different batch sizes or on different datasets?
>
> A4: Thank you for raising this interesting point. We revised the manuscript and provided the learning curves in Figure 14 in the Supplementary. Different batch sizes tend to have a similar trend of convergence.
>
> ----
>
> **Q5**: What is the impact of local epochs on the proposed method?
>
> A5: We provided analysis on the impact of local epochs in Table 14 in the Supplementary. Our proposed FedWon is robust on different local epochs and consistently outperforms FedBN under different settings.

---

### Author Response · Authors · 2023-11-17
**Rebuttal Summary**

We sincerely thank all reviewers for their valuable comments and suggestions. We have made the following updates (where + denotes the newly added content during rebuttal and * denotes modified content based on the initial submission.):

*Section 2.1: added discussions of related works: PartialFed and FMTDA.

*Figure 1b: revised to compare BN stats of two clients only.

*Figure 12: separated the visualization of BN statistics between clients and the server into multiple figures for clarity.

*Figure 4 (left): revised the results of batch size B = 2.

+Table 7: summary of compared methods on different aspects.

+Table 16: added new experimental results on cross-device FL with a total of 1000 clients.

+Table 17: added new experimental results of FedWon and PartialFed.

+Figure 14: added experimental results of convergence of different batch sizes.


We have revised our paper according to all the valuable comments and please let us know if there is anything still not clear. We are happy to run more experiments if you have further suggestions.

---

### Meta-Review · Area_Chair_UZFa · 2023-12-13

**Metareview:**

This paper proposes a versatile method for multi-domain federated learning. It has been assessed by four knowledgeable reviewers who split their scores. Two scored it as acceptable (one full accept, one marginal accept) and two as rejectable (both marginal rejects).  The paper includes comprehensive experimental evaluation of the proposed method that demonstrates its potential utility. The reviewers, however, voiced their legitimate concerns about the novelty of the approach, and point out that its utility is limited to CNNs. All things considered, this paper can be of interest to the ICLR audience, and so even though its current form is just marginally above the acceptance threshold I very weakly lean towards recommending its acceptance.

**Justification For Why Not Higher Score:**

This paper is marginally acceptable.

**Justification For Why Not Lower Score:**

n/a

---

### Decision · Program_Chairs · 2024-01-16

Accept (poster)